# Structural and functional characterization of cyclic pyrimidine-regulated anti-phage system

Mei-Hui Hou[1,4], Chao-Jung Chen [2,3,4], Chia-Shin Yang[1], Yu-Chuan Wang [1] & Yeh Chen [1] ✉

3′,5′-cyclic uridine monophosphate (cUMP) and 3′,5′-cyclic cytidine monophosphate (cCMP) have been established as bacterial second messengers in the phage defense system, named pyrimidine cyclase system for anti-phage resistance (Pycsar). This system consists of a pyrimidine cyclase and a cyclic pyrimidine receptor protein. However, the molecular mechanism underlying cyclic pyrimidine synthesis and recognition remains unclear. Herein, we determine the crystal structures of a uridylate cyclase and a cytidylate cyclase, revealing the conserved residues for cUMP and cCMP production, respectively. In addition, a distinct zinc-finger motif of the uridylate cyclase is identified to confer substantial resistance against phage infections. Furthermore, structural characterization of cUMP receptor protein PycTIR provides clear picture of specific cUMP recognition and identifies a conserved N-terminal extension that mediates PycTIR oligomerization and activation. Overall, our results contribute to the understanding of cyclic pyrimidine-mediated bacterial defense.

Billions of years of coevolution between bacteria and phages have led to the development of more than 100 bacterial defense systems against phage infections[1]. Meanwhile, phages have evolved proteins to counteract bacterial defenses, thus allowing phage survival. As a consequence, nucleotide signal-mediated bacterial defense systems evolved, which are highly diverse in their responses, giving bacteria survival advantage[2]. Purine nucleotide signals have been subject to research for decades. In contrast, pyrimidine-containing nucleotide signals, including cyclic mono-pyrimidines[3] and cyclic di-pyrimidines[4,5], have only recently been discovered and appreciated.

Owing to the advances in mass spectrometry, 3′,5′-cyclic uridine monophosphate (cUMP) and 3′,5′-cyclic cytidine monophosphate (cCMP) have been detected in numerous animal species and cell types[6,7]. The biological function of cUMP is associated with apoptosis and necrosis[8,9], whereas cCMP is linked to vasodilation, platelet aggregation inhibition, immune modulation, and embryonic development[10,11]. The highest concentration of cUMP and cCMP was detected in astrocytes[12], which account for 85% of brain cells[13]. Astrocyte dysfunction causes neurodegenerative diseases[14] as well as glioblastoma, a type of brain cancer[15]. Therefore, understanding the signaling mechanism mediated by cUMP and cCMP may facilitate the development of therapeutics for strokes, brain tumors, and neurodegenerative diseases[16]. In addition, the bacterial toxins edema factor and CyaA secreted by *Bacillus anthracis* and *Bordetella pertussis*, respectively, can synthesize cUMP, and cCMP[17]. These enzymes inhibit the synthesis of thromboxane B2 and leukotriene B4 from macrophages, thereby attenuating inflammatory responses of host cells[18]. These results suggest the important roles played by cUMP and cCMP in mammalian immune systems. However, due to the base-promiscuity of mammalian nucleotide cyclases and bacterial toxins, it is still unclear whether cyclic mono-pyrimidines serve as signaling molecules or just by-products.

[1]Department of Food Science and Biotechnology, National Chung Hsing University, Taichung 40227, Taiwan. [2]Graduate Institute of Integrated Medicine, China Medical University, Taichung 40447, Taiwan. [3]Proteomics Core Laboratory, Department of Medical Research, China Medical University Hospital, Taichung 40447, Taiwan. [4]These authors contributed equally: Mei-Hui Hou, Chao-Jung Chen. ✉e-mail: chyeah6599@nchu.edu.tw

Until 2021, scientists reported that a group of bacterial nucleotide cyclases, which utilize UTP, and CTP to synthesize cUMP and cCMP, defend against bacteriophage infection, proving that cUMP, and cCMP are authentic second messengers controlling anti-phage defense in prokaryotes[3]. This anti-phage system, termed pyrimidine cyclase system for antiphage resistance (Pycsar), contains a pyrimidine cyclase PycC and a receptor protein with either transmembrane domain (PycTM) or Toll/interleukin-1 receptor domain (PycTIR)[3]. Upon phage infections, produced cyclic mono-pyrimidines activates their receptors PycTM, which disrupts membrane integrity by forming transmembrane-spanning pore, or PycTIR, which deplete NAD⁺, leading to growth arrests and cell deaths, thus protecting the uninfected bacterial populations, a strategy called "abortive infection"[19]. Pyrimidine cyclase PycCs share conserved catalytic motifs with class III adenylate and guanylate cyclase enzymes and require divalent cations, such as $Mg^{2+}$ or $Mn^{2+}$ for catalysis[20]. Based on phylogenetic analysis, pyrimidine cyclase PycCs were assigned to five subclades from clade A to clade E. Clade A PycC cyclases have two consecutive nucleotide cyclase domains, whereas clade B–E cyclases contains only one nucleotide cyclase domain[3]. However, due to the limited structural information of PycC cyclases, the molecular mechanisms underlying the synthesis of cUMP and cCMP remain unclear. In Pycsar defense system, the receptor PycTIR is composed of a N-terminal cyclic nucleotide-binding domain (CNBD) and a C-terminal TIR domain. Binding of cUMP to its N-terminal CNBD promotes the oligomerization of PycTIR proteins, followed by the activation of its NADase activity of the C-terminal TIR domain. Depletion of the essential cofactor NAD⁺ by PycTIR proteins will lead to growth arrests and even cell death of the bacterial hosts, stopping the phage replication and propagation. Previous study has shown that the NAD⁺ cleavage activity of PycTIR proteins could be stimulated by little amount of cUMP (50–100 nM); in contrast, other cyclic mononucleotides, even at concentrations up to 1 mM, still cannot activate PycTIR proteins[3]. Furthermore, in vivo phage challenge assays demonstrated that ectopic addition of cUMP activated PycTIR-expressing E. coli cells for phage defense, whereas the addition of cCMP did not. All these results showed the specificity of the CNBD of PycTIR to cUMP[3]. However, the detailed molecular mechanism underlying specific cUMP recognition and oligomerization by PycTIR proteins remains poorly understood.

In this study, we aimed to structurally, and functionally characterize both cUMP- and cCMP-synthesizing PycC and cUMP receptor PycTIR. Our results provide important information regarding pyrimidine substrate recognition by PycC enzymes and demonstrate the specific binding mechanism of cUMP to PycTIR, contributing to our understanding of the biological functions of cyclic pyrimidines.

## Results

### Crystal structure of a clade E PycC cyclase EaPycC

Among the discovered pyrimidine cyclases, only clade E PycC cyclase can synthesize cCMP products. However, there is no crystal structure of clade E PycC cyclase available. To understand the molecular mechanism underlying the synthesis of cCMP, the cytidylate cyclase activity of a clade E PycC from multidrug-resistant bacterium *Elizabethkingia anophelis*[21] (*Ea*PycC) was first biochemically characterized. By incubating purified *Ea*PycC proteins with CTP substrates, an elution peak corresponding to the cCMP product was detected, according to the matched LC retention time and MS/MS spectra with the chemical standard (LC-MS dataset, Supplementary Fig. 1a, b). The native crystals of *Ea*PycC that belongs to space group of *I*23 were subsequently obtained, which contains two *Ea*PycC molecules in the asymmetric unit. The structure of *Ea*PycC was solved by molecular replacement and refined to 2.2-Å resolution (Supplementary Table 1). Each *Ea*PycC molecule has a central seven-stranded β-sheet wrapped around three main α-helices (Fig. 1a). *Ea*PycC assembles into a head-to-tail homodimer with two symmetric active sites at the dimer interface (Fig. 1b).

Previous studies have determined structures of a clade A PycC from *Pseudomonas aeruginosa* (*Pa*PycC, PDB: 6YII) and a clade B PycC from *Burkholderia cepacia* LK29 (*Bc*PycC, PDB: 7R65[3]). Structural comparative analysis showed that *Ea*PycC share similar nucleotide cyclase core domain with both *Pa*PycC and *Bc*PycC with root-mean-square deviations (RMSDs) of 3.7 Å and 2.7 Å for 205 and 186 matched Cα pairs, respectively (Supplementary Fig. 2a, b). The N-terminal α helix (α7') previously identified to be a hallmark of PycC cyclases, is also present in *Ea*PycC (Fig. 1c–e). However, unlike *Bc*PycC, the N-terminal α helix (α1) of *Ea*PycC packs against α4 and α5 helices in the same protomer instead of extending into the opposing protomer (Fig. 1c). Another distinct feature found in *Ea*PycC is a C-terminal Y-shaped structure that comprises two anti-parallel β-sheets (β10–β13) near the active sites (Fig. 1c, d). In contrast, *Bc*PycC only has a simpler β-hairpin structure at the equivalent position (Fig. 1c, d). Similarly, the C-terminal Y-shaped structure of *Ea*PycC is also absent in *Pa*PycC structure (Fig. 1e). The active site of *Ea*PycC contains two conserved catalytic residues D102 and D146 responsible for divalent metal coordination[20], which superimposed well with those in *Pa*PycC and *Bc*PycC (Supplementary Fig. 2c). In contrast to the UTP-recognition residues, Y50, D94, and R97, identified in *Bc*PycC, *Ea*PycC contains strikingly different specificity-determining residues, F100, R142, and Q144, for CTP binding (Supplementary Fig. 2d). Indeed, mutating F100, R142, and Q144 to alanine residues abolished the cytidylate cyclase activity of *Ea*PycC, yielding no cCMP products (LC-MS dataset, Supplementary Fig. 1c).

### Crystal structure of a clade D PycC cyclase AnPycC

PycC cyclases from clades A–D are known to convert UTP to cUMP. Currently, only structures of clade A and B PycC without bound pyrimidine nucleotides have been determined. How PycC cyclases specifically recognize UTP substrates and catalyze the synthesis of cUMP from UTP are still ambiguous. Here, a clade D PycC from *Anabaena sp.* (*An*PycC) was chosen for structural and functional characterization. The production of cUMP by purified *An*PycC proteins was confirmed by LC-MS/MS compared with the chemical standard (LC-MS dataset, Supplementary Fig. 3a–d). The crystal structure of *An*PycC was subsequently determined to 1.8-Å resolution (Supplementary Table 2). *An*PycC contains two nearly identical polypeptides with an r.m.s.d. of 0.13 Å for 259 Cα pairs, two chloride ions (Cl⁻), two zinc ions (Zn²⁺), two acetate groups, and two glycerol molecules in the asymmetric units. Each *An*PycC polypeptide core domain has a central seven-stranded β-sheet surrounded by α-helices (Fig. 2a). *An*PycC forms a tight, head-to-tail homodimer with catalytic centers located at the dimerization interface, with a buried surface area of approximately 3,700 Å² (Fig. 2b). The glycerol molecules form hydrogen bonds with K64, S195, and K246 (Fig. 2c). The zinc ions are coordinated to the C221, C223, C260, and H263 sidechains on the surface of *An*PycC (Fig. 2d). Notably, these four zinc-binding residues are highly conserved among the clade D PycC cyclases (Fig. 2e), but not clade A, B, C, and E PycC cyclases. To test whether these zinc-binding residues are involved in cUMP production by *An*PycC, and abortive infection phenotypes mediated by *Anabaena sp.* Pycsar system containing cUMP-synthesizing *An*PycC and cUMP receptor *An*PycTM, we made a triple alanine mutant of *An*PycC (C221A/C223A/C260A). In vitro enzyme activity assay showed that no cUMP products were detected in the overnight reaction containing *An*PycC^C221A/C223A/C260A mutant proteins and UTP substrates, suggesting the importance of zinc-binding residues for the uridylyl cyclase activity of *An*PycC (LC-MS dataset, Supplementary Fig. 3e). In vivo toxicity analysis in E. coli demonstrated that E. coli cells expressing the *An*Pycsar system elicited severe growth inhibition and cell death compared to control cells without IPTG induction (Fig. 2f). Mutating the zinc-binding residues of *An*PycC to alanine restored the growth of E. coli cells close to control cells (Fig. 2f). Phage infection experiments further showed that E. coli cells harboring wild-type *An*Pycsar system conferred significant resistance to invading phages but completely

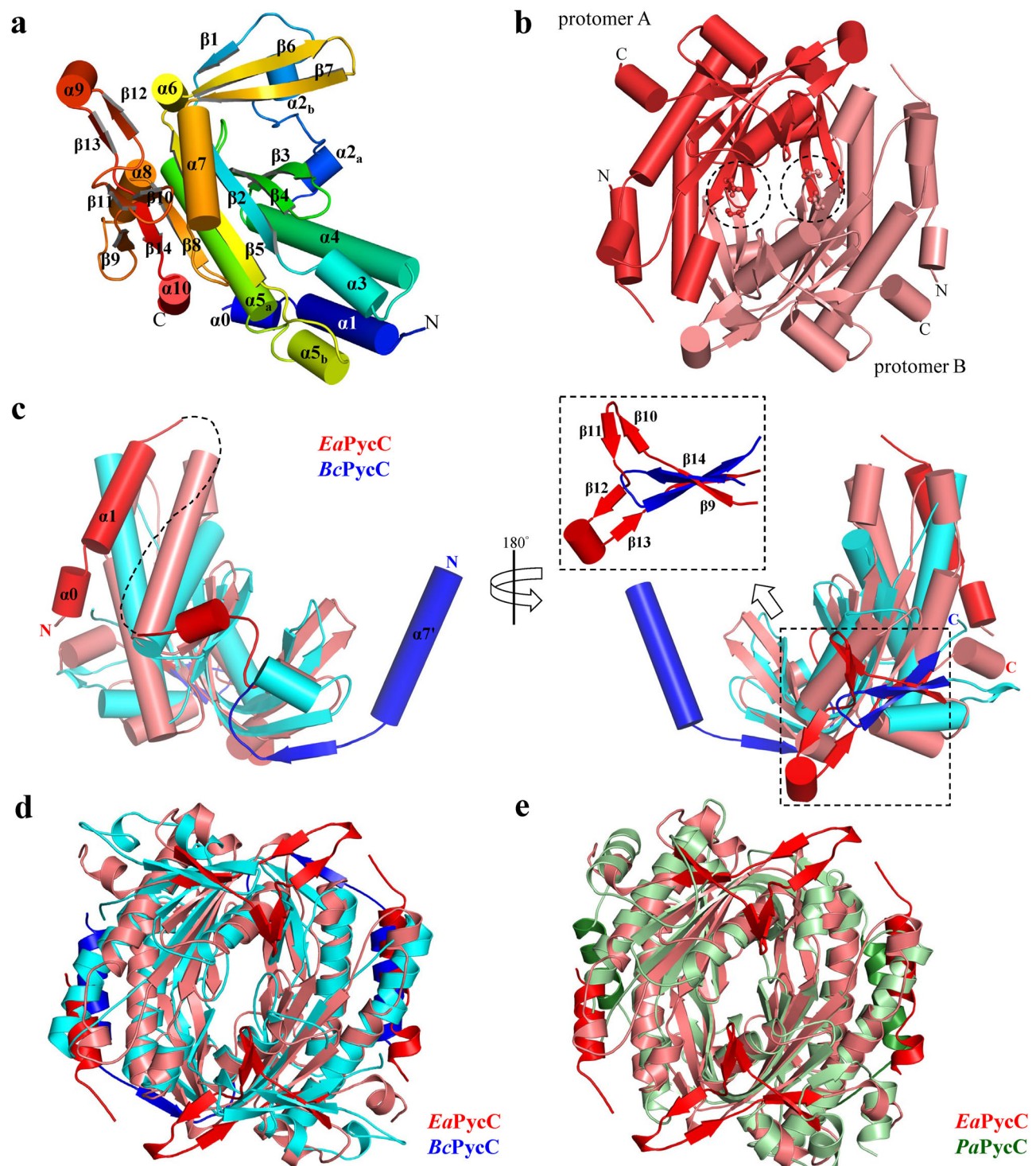

**Fig. 1 | Structural characterization of a cCMP-synthesizing PycC cyclase,** ***Ea*PycC. a**, **b** Cartoon representation of (**a**) protomer and (**b**) dimer structure of *Ea*PycC. The N- and C-terminus are labeled. The symmetric active sites of *Ea*PycC dimer are indicated by black dashed circles and the conserved catalytic residues D102 and D146 are shown in ball-and-stick model. **c** The front (left) and back (right) view of the superimposed *Ea*PycC and *Bc*PycC promoters. The missing residues

(a.a. 22–66) of *Ea*PycC protomer are indicated by black dashed lines. The inlet shows the enlarged view of the superimposed Y-shaped structure of *Ea*PycC with β-hairpin of *Bc*PycC at the equivalent position. **d**, **e** Structural alignment of *Ea*PycC dimer with (**d**) *Bc*PycC dimer, and (**e**) *Pa*PycC dimer. *Ea*PycC, *Bc*PycC, and *Pa*PycC proteins are colored in salmon, cyan, and pale-green, with their distinct N- and C-terminal structural elements colored in red, blue, and forest-green respectively.

lose the anti-phage activity by mutating the zinc-binding residues of *An*PycC (Fig. 2g). Altogether, these data support the important role played by the zinc-binding residues for cUMP production by *An*PycC proteins and phage resistance mediated by the *An*Pycsar system.

Superimposition of ATP-bound adenylyl cyclase (PDB: 1WC1[22]) with *An*PycC revealed that the active sites of *An*PycC are too small to accommodate nucleotide substrates, suggesting that the *An*PycC structure presented here is in its inactive state conformation

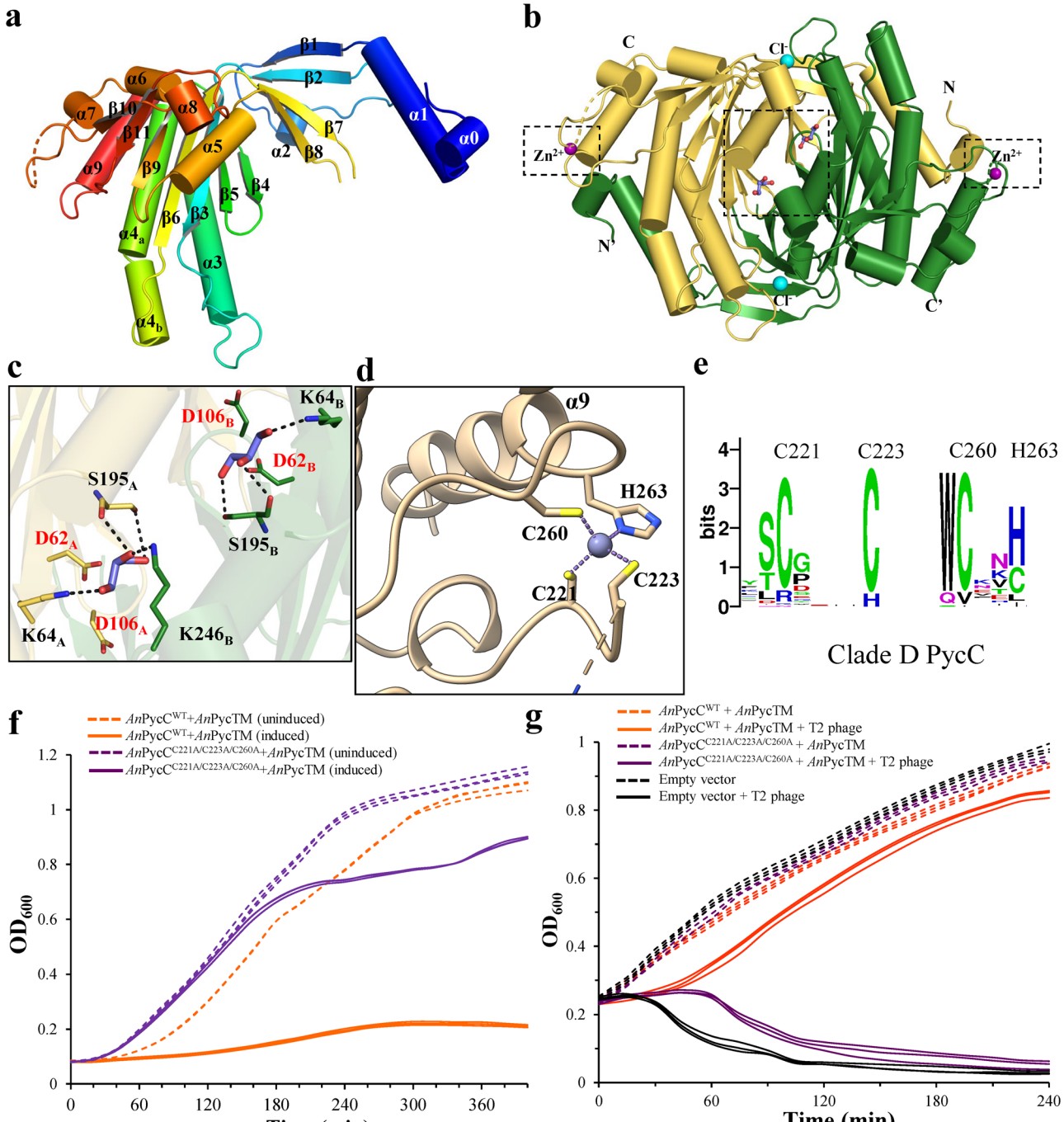

**Fig. 2 | Clade D PycC contains a conserved zinc-finger, which regulates uridylate cyclase and anti-phage ability. a**, **b** Cartoon representation of (**a**) protomer and (**b**) dimer structure of *An*PycC. The N- and C-terminus are labeled. The bound zinc and chloride ions are shown in spheres and indicated. The two glycerol molecules bound at the active sites are shown in sticks. **c** The detailed interaction between the bound glycerol and residues from *An*PycC. The residues and the glycerols are shown in sticks. The H-bonds are shown in black dashed lines. The conserved catalytic residues, D62 and D106, required for divalent cation coordination located away from the bound glycerols. **d** The zinc-finger domain of *An*PycC. The zinc ion is tetrahedrally coordinated by three cysteine residues C221, C223, and C260, and one histidine residue H263. **e** The sequence logo of zinc-finger domain in clade D uridylate cyclases. The zinc-binding residues of *An*PycC are indicated above. **f** The

growth curves of *E. coli* cells overexpressing wild-type *An*PycC and its effector *An*PycTM (orange lines) and *E. coli* cells overexpressing *An*PycC triple mutant (C221A/C223A/C260A) and *An*PycTM (purple lines) compared with uninduced control cells (dashed lines). Source data are provided as a Source Data file. **g** The growth curves of *E. coli* cells overexpressing wild-type *An*PycC and *An*PycTM (orange lines), *E. coli* cells overexpressing *An*PycC triple mutant (C221A/C223A/C260A) and *An*PycTM (purple lines), and *E. coli* cells harboring empty vector (black lines) with T2 phage infection at an MOI of 0.1. The *E. coli* cells harboring *An*Pycsar system or empty vector without T2 phage infection and protein induction serve as negative controls (dashed lines). Source data are provided as a Source Data file. The experiments in (**f**) and (**g**) were performed for *n* = 3 biological replicates and each of them is shown.

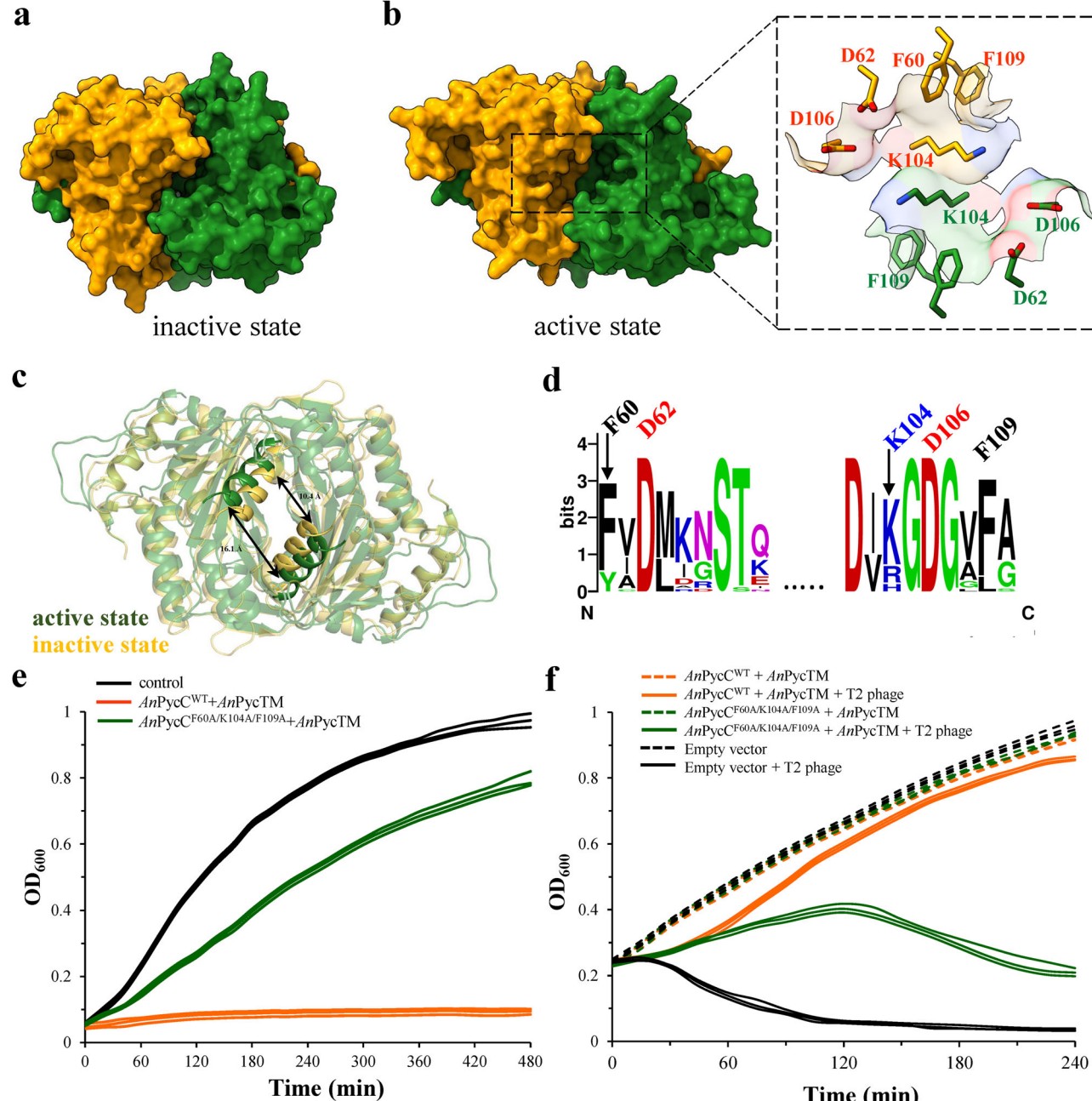

**Fig. 3 | Clade D PycC utilized conserved bulky residues, F60, K104, and F109 for UTP substrate recognition. a** Surface presentation of the crystal structure of *An*PycC in its inactive state conformation. **b** Left, surface presentation of predicted *An*PycC in its active state conformation. Right, the enlarged view of the active site of *An*PycC, showing the conserved residues (in sticks) lining the binding pockets. **c** Structural comparison of inactive and active *An*PycC dimers. The distance between the α5 helices of *An*PycC protomers are indicated. **d** The sequence logo of active-site residues in clade D uridylate cyclases. The residues in *An*PycC are indicated above. **e** The growth curves of *E. coli* cells overexpressing wild-type *An*PycC and its effector *An*PycTM (orange), *E. coli* cells overexpressing *An*PycC triple

mutant (F60A/K104/F109A) and *An*PycTM (green), and control cells with empty vector (black). Source data are provided as a Source Data file. **f** The growth curves of *E. coli* cells overexpressing wild-type *An*PycC and *An*PycTM (orange), *E. coli* cells overexpressing *An*PycC triple mutant (F60A/K104A/F109A) and *An*PycTM (green), and control cells (black) with T2 phage infection at an MOI of 0.1. The *E. coli* cells harboring *An*Pycsar system or empty vector without T2 phage infection and protein induction serve as negative controls. Source data are provided as a Source Data file. The experiments in (**e**) and (**f**) were performed for *n* = 3 biological replicates and each of them is shown.

(Fig. 3a and Supplementary Fig. 4). To further explore different conformational states of *An*PycC, structural prediction software AlphaFold2[23] was used. One resulting *An*PycC model showed larger active sites than that of the inactive *An*PycC structure (16.1 Å vs 10.4 Å, Fig. 3c). Furthermore, its catalytic aspartate residues are superimposed well with those of *Pa*PycC, *Bc*PycC, and *Ea*PycC and point toward the active sites (Fig. 3b and Supplementary Fig. 2c),

suggesting this predicted *An*PycC model is in its active state conformation. Importantly, in contrast to previously identified residues, Y50, D94, and R97, of clade B *Bc*PycC for pyrimidine base selection, it is found that F60, K104, and F109 are located at the nucleotide-binding pocket of active *An*PycC and are highly conserved among clade D PycC cyclases (Fig. 3b, d). To investigate if these residues are also critical for UTP recognition and thus the downstream effector

activation, we changed F60, K104, and F109 of *An*PycC to alanine residues. As expected, no cUMP products were detected by including UTP substates and *An*PycC^F60A/K104A/F109A mutant proteins (LC-MS dataset, Supplementary Fig. 3f). Moreover, *E. coli* cells expressing *An*PycC^F60A/K104A/F109A triple mutant and effector *An*PycTM exhibited remarkably reduced toxicity than *E. coli* cells expressing wild-type *An*Pycsar system (Fig. 3e). Phage infections in *E. coli* liquid cultures further showed that *E. coli* cells expressing *An*PycC^F60A/K104A/F109A triple mutant collapsed in contrast to the normal growth of cells expressing *An*PycC^WT protein (Fig. 3f). Altogether, these data proved the

importance of the conserved active-site residues F60, K104, and F109 of the clade D *An*PycC cyclase for cUMP production and phage resistance.

Comparative structural analysis demonstrated that the structure of clade B *Bc*PycC was most similar to that of *An*PycC with an r.m.s.d. of 2.7 Å for 223 Cα pairs. The N-terminal α-helices, α0 and α1, of *An*PycC make extensive contacts with α3 and α4 of the opposing protomer, similar to those of *Bc*PycC (Figs. 2a, b and 4a, b). Notably, *An*PycC has an additional C-terminal α-helix α9, named zinc-binding ribbon, which further interacts with α4 and the N-terminus of the opposing protomer

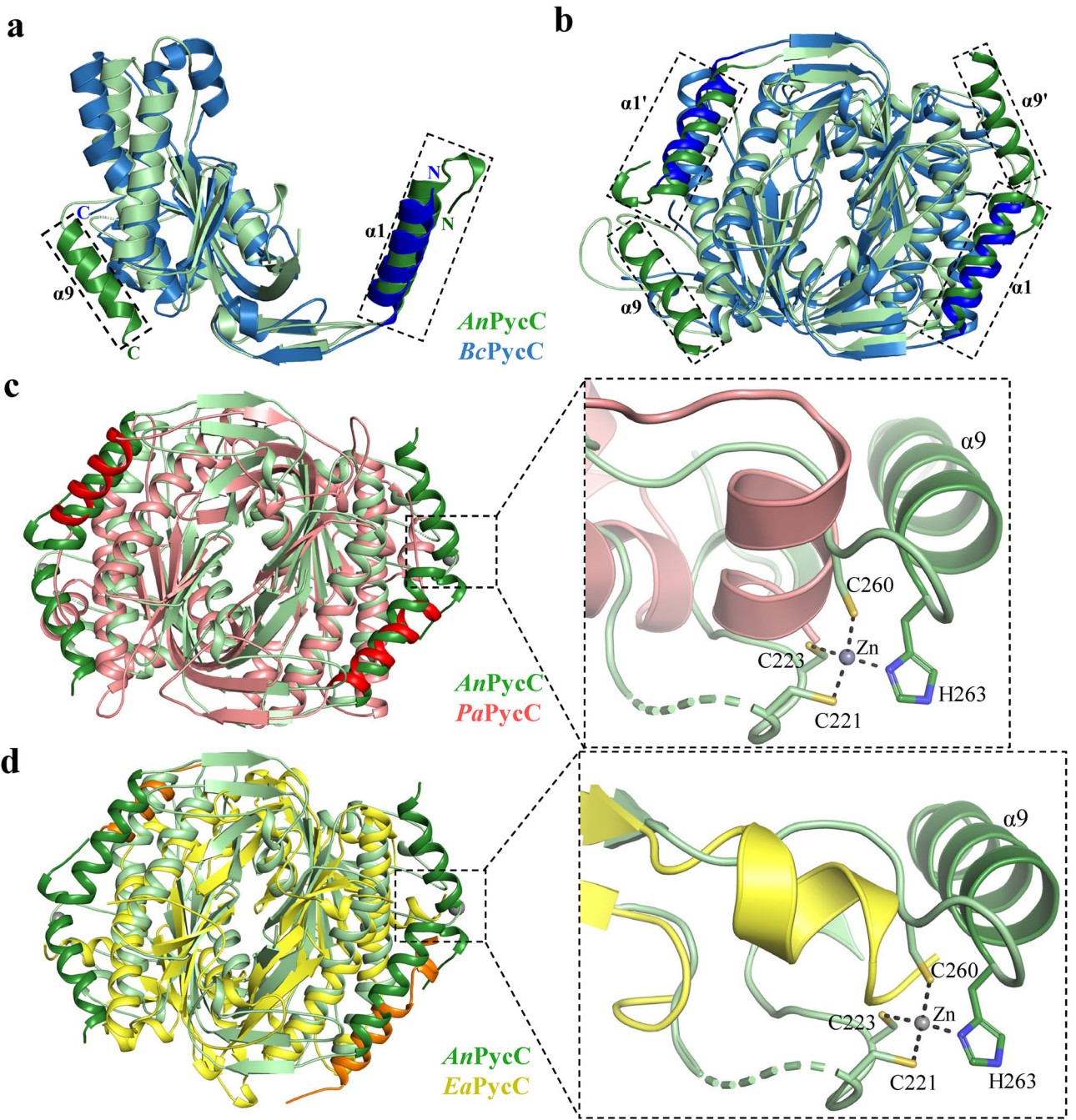

**Fig. 4 | Structural comparison of *An*PycC with other PycC pyrimidine cyclases.** **a** Structural comparison of *An*PycC protomer with *Bc*PycC protomer. **b** Structural comparison of *An*PycC dimer with *Bc*PycC dimer. **c** Left, superimposed *An*PycC dimer with *Pa*PycC. Right, the enlarged view of the superimposed C-terminal structures of *An*PycC and *Pa*PycC. **d** Left, superimposed *An*PycC dimer with *Ea*PycC dimer. Right, the enlarged view of the superimposed C-terminal structures of

*An*PycC and *Ea*PycC. *An*PycC, *Bc*PycC, *Pa*PycC, and *Ea*PycC proteins are colored in pale-green, sky-blue, salmon, and yellow with their N- and C-terminal structural elements colored in forest-green, blue, red, and orange respectively. The zinc-binding residues of *An*PycC located at α9 helix and β10-α7 loop are shown in sticks and the bound zinc ions are shown in grey spheres.

(Figs. 2d and 4a, b). In contrast, clade A *Pa*PycC exhibited large structural differences when compared with *An*PycC with an r.m.s.d. of 3.4 Å for 196 Cα pairs. Both *An*PycC and *Pa*PycC have extra α-helices at the N-terminus (Fig. 4c). However, *An*PycC has an additional zinc-binding ribbon at the C-terminus, unlike the short helix linker of *Pa*PycC at the equivalent position (Fig. 4c). Similarly, *Ea*PycC contains only a short C-terminal helix instead of the zinc-binding ribbon of *An*PycC (Fig. 4d).

## Specific recognition of cUMP by PycTIR

PycTIR consists of an N-terminal cyclic nucleotide-binding domain (CNBD) and a C-terminal TIR domain that is linked to the cUMP-synthesizing PycC in genomic contexts[3]. To determine how cyclic pyrimidines are specifically recognized, the N-terminal CNBD (a.a. 1–150) of the PycTIR from *Novosphingobium pentaromativorans* (*Np*PycTIR$_{CNBD}$), *Pseudovibrio* sp. (*Ps*PycTIR$_{CNBD}$), *Acinetobacter haemolyticus* (*Ah*PycTIR$_{CNBD}$), and *Stenotrophomonas maltophilia* (*Sm*PycTIR$_{CNBD}$) were cloned, expressed, and purified. ITC results showed that both *Np*PycTIR$_{CNBD}$ and *Ps*PycTIR$_{CNBD}$ bind tightly to cUMP with nanomolar affinity at a 1:1 binding stoichiometry (Table 1 and Supplementary Fig. 5a, b). cUMP also showed strong binding to *Ah*PycTIR$_{CNBD}$ and *Sm*PycTIR$_{CNBD}$ with dissociation constant ($K_D$) of 8.2 × 10$^{-7}$ M and 2.7 × 10$^{-7}$ M (Table 1 and Supplementary Fig. 5c, d), demonstrating that PycTIR is a high-affinity cUMP receptor. Moreover, cUMP caused the most significant shift of 21 °C in comparison to other cyclic mononucleotides in the protein thermal shift assay, further supporting this notion (Supplementary Fig. 6).

Subsequently, we determined the crystal structures of *Np*PycTIR$_{CNBD}$ and *Ps*PycTIR$_{CNBD}$ complexed with cUMP at 2.86- and 2.4-Å resolution, respectively (Supplementary Tables 3 and 4). The *Np*PycTIR$_{CNBD}$ crystal structure in the space group $P6_522$ contains only one polypeptide in the asymmetric unit (from residues 1–91 and 98–140), forming a central six-stranded antiparallel β-barrel surrounded by five α-helices (Fig. 5a). The loop connecting β5 and β6 and the last ten amino acids at the C-terminus are unresolved in the *Np*PycTIR$_{CNBD}$ crystal structure owing to structural flexibility. The structure of *Ps*PycTIR$_{CNBD}$–cUMP, solved in the *I*4 space group, contains two continuous polypeptides from residue 1–150 and the C-terminal His$_6$-tag, which can assemble into a dimer (Fig. 5b), similar to the known cAMP receptor protein (CRP)[24,25]. In the complex structure, we observed a clear and strong electron density corresponding to cUMP within the β-barrel (Supplementary Fig. 7a). Structural comparison of *Np*PycTIR$_{CNBD}$ with *Ps*PycTIR$_{CNBD}$ shows r.m.s.d. of 1.1 and 0.9 for 116 and 117 Cα pairs, respectively. Binding of cUMP stabilizes the ligand-binding pocket, promoting the formation of an additional short α4 helix (Fig. 5c). cUMP also causes a large structural movement of the C-terminal stem helix toward the ligand-binding pocket, producing dimerization of *Ps*PycTIR$_{CNBD}$ (Fig. 5d). The dimerization of *Ps*PycTIR$_{CNBD}$ is mediated through hydrophobic interactions, including residues Y128, K129, I131, A132, L135, A136, and L139 (Fig. 5d).

Specific recognition of cUMP by *Ps*PycTIR is dictated by the N143 sidechain that forms two hydrogen bonds with the N$_3$ and O$_4$ atoms and the S98 residue that forms one hydrogen bond with the O$_2$ atom of the uracil base (Fig. 6a). Furthermore, the sidechain of R138 forms extensive cation–π stacking interactions with the uracil base of the

### Table 1 | A summary of dissociation constants of different PycTIR proteins for cUMP as determined by isothermal titration calorimetry

| PycTIR proteins | $K_D$ (M) |
| --- | --- |
| *Ps*PycTIR$_{CNBD}$ | 1.0 × 10$^{-9}$ |
| *Np*PycTIR$_{CNBD}$ | 1.0 × 10$^{-9}$ |
| *Ah*PycTIR$_{CNBD}$ | 8.2 × 10$^{-7}$ |
| *Sm*PycTIR$_{CNBD}$ | 2.7 × 10$^{-7}$ |

bound cUMP (Fig. 6a). The cUMP ribose 2′-OH hydrogen-bonded to the side and main chains of E87 and G86, respectively, and is located at the α4 helix, whereas the 3′,5′-phosphodiester linkage is recognized by A89, R97, and S98 (Fig. 6a). Additionally, *Ps*PycTIR forms hydrophobic interactions with the bound cUMP via I50, I69, I76, I85, A99, and V101 (Supplementary Fig. 7b). A comparative structural study showed that the specificity-determining residue N143 of *Ps*PycTIR for uracil base recognition is located in the same position as the residue S128 of *E. coli* CRP (Fig. 6b), which is responsible for adenine base recognition[24,25]. Multiple sequence alignments of CNBD-containing proteins revealed that G86, E87, and R97 of *Ps*PycTIR are highly conserved (Fig. 6c) and have similar roles in both ribose 2′-OH and 3′,5′-phosphodiester linkage recognition[26]. In contrast, the identified N143 of *Ps*PycTIR is conserved only in PycTIR proteins, indicating its potential role in the specific binding of cUMP (Fig. 6c). To validate the functional importance of the cUMP-binding residues identified in the complex structure, we constructed single *Ps*PycTIR mutants, including R138A, N143A, and N143S. Enzyme activity assays using the fluorescent NAD$^+$ analog, ε-NAD, revealed that *Ps*PycTIR protein was fully activated by cUMP at a concentration of 2.5 μM cUMP (Fig. 6d). In contrast, addition of up to 10 μM cAMP, cGMP or cCMP did not activate the NADase activity of *Ps*PycTIR, indicating that it is a cUMP-specific effector (Fig. 6e). Changing the cUMP recognition residues, R138 and N143, to alanine or serine, rendered *Ps*PycTIR insensitive to the presence of activating signal cUMP (Fig. 6f), supporting their roles for dictating cUMP binding. In liquid culture, *E. coli* expressing wild-type *Ps*PycTIR (*Ps*PycTIR$^{WT}$) exhibited severe growth inhibition upon addition of cUMP, indicating strong in vivo NADase activity of *Ps*PycTIR stimulated by cUMP, leading to NAD$^+$ depletion (Fig. 6g). Notably, even in the presence of high cUMP concentrations, *E. coli* expressing single mutants of *Ps*PycTIR, *Ps*PycTIR$^{R138A}$, and *Ps*PycTIR$^{N143A}$ exhibited restored bacterial growth compared with that of control cells (Fig. 6g). In phage infection experiments, *E. coli* cells expressing wild-type *Ps*Pycsar system showed continuous growth regardless of T2 phage infections, indicating *Ps*Pycsar system provide strong defense against T2 phage (Fig. 6h). In contrast, *E. coli* cells harboring empty vector rapidly collapsed after 1-hour post-infection (Fig. 6h). Moreover, *Ps*Pycsar-expressing cells containing *Ps*PycTIR$^{N143A}$ mutant are much more susceptible to phage infection than *E. coli* cells harboring wild-type *Ps*Pycsar system (Fig. 6h), validating the functional importance of cUMP-recognition residues for TIR activation.

## cUMP-mediated oligomerization and activation of PycTIR

Protein oligomerization has been known to activate several TIR domain-containing effectors for anti-phage defense[27–29]. However, the detailed molecular mechanism underlying PycTIR oligomerization is unknown. Here, we found that *Ps*PycTIR can oligomerize. The cUMP-bound *Ps*PycTIR$_{CNBD}$ dimers pack adjacently to form continuous linear filaments (Fig. 7a). The *Ps*PycTIR$_{CNBD}$ dimer-to-dimer interface ( ~ 1,100 Å$^2$) comprises interactions between the A-to-A′, A-to-B′, and B-to-B′ protomers (Fig. 7b). The extensive A-to-A′ (or B-to-B′) protomer interface consists of one salt-bridge (R12-E114), four hydrogen bonds (S64-K96, M1-E55, R4-E55, and R4-P93), and numerous hydrophobic interactions between the residues M1, R4, F5, R12, L13, M20, I63, and S64 from α1, α2, and β2 of protomer A (B) and E55, N57, P93, T94, K96, E114, A115, and S118 from α4′, α5′, and α6′ of protomer A′ (B′) (Fig. 7c, d and Supplementary Fig. 8a). The cross-interaction between the protomers A and B′ constitutes five inversely arranged residues E16, M20, R25, G26, and K129 from both protomers, where E16 and K129 formed two symmetric salt-bridges (Supplementary Fig. 8b). Protein structure similarity search[30] revealed that the structure of a hyperpolarization-activated cyclic nucleotide-modulated (HCN) ion channel from sea urchin *Strongylocentrotus purpuratus* (PDB: 2PTM[31]) was most similar to *Ps*PycTIR$_{CNBD}$ with an r.m.s.d. of 2.2 Å for 132 Cα pairs. Each promoter of the HCN channel contains six α-helices at the N-terminus of

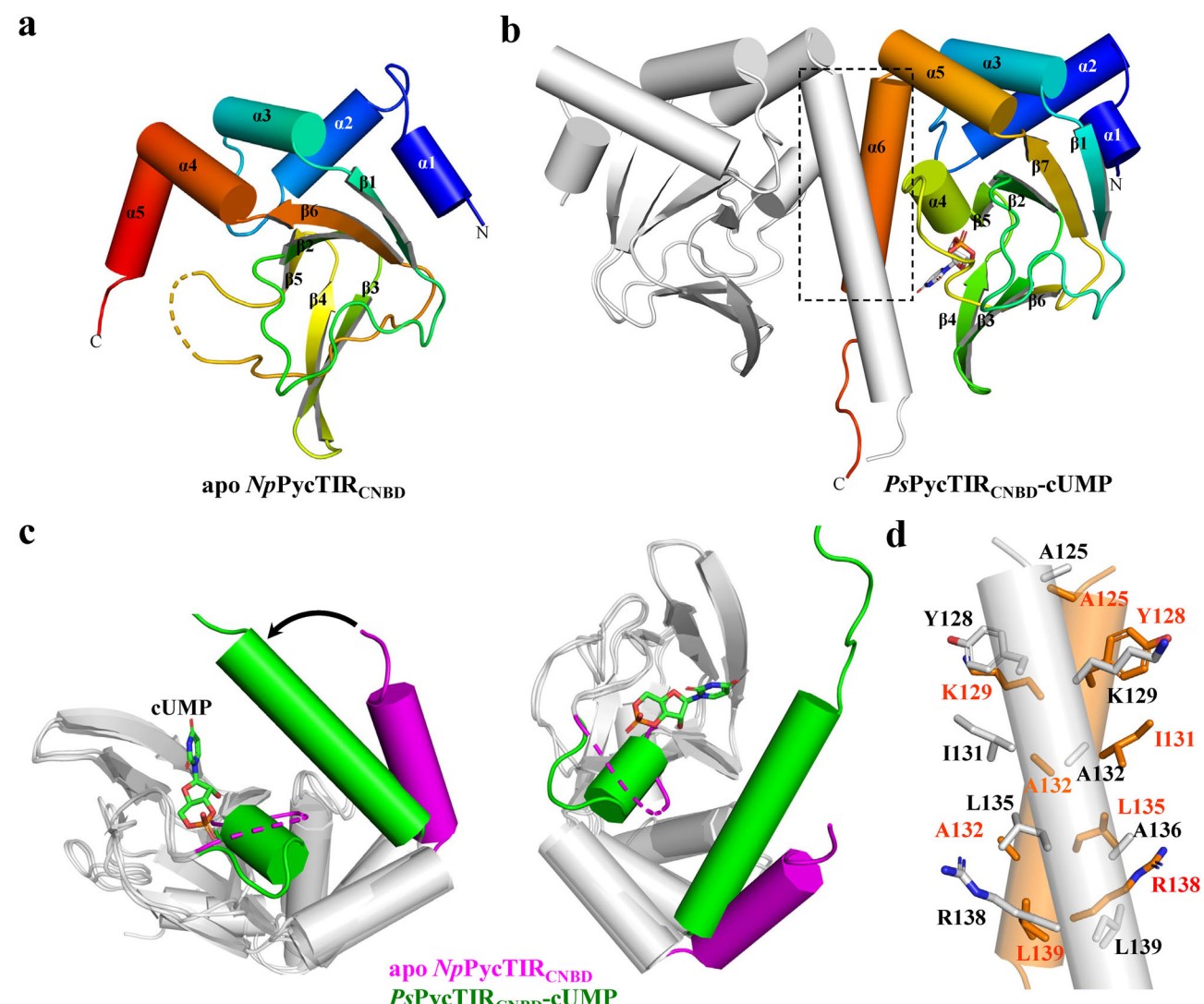

**Fig. 5 | Structural characterization of cUMP receptor PycTIR proteins. a** Crystal structure of apo $Np$PycTIR$_{CNBD}$ with five α-helices and six β-strands indicated. **b** Crystal structure of dimeric $Ps$PycTIR$_{CNBD}$ complexed with cUMP. Each protomer consists of six α-helices and seven β-strands. The bound cUMP is shown in stick. The dimerization interface is highlighted. **c** Structural comparison of apo $Np$PycTIR$_{CNBD}$ (magenta) with cUMP-bound $Ps$PycTIR$_{CNBD}$ (green). The major structural differences between them are colored. **d** The CNBD dimerization interface of cUMP-bound $Ps$PycTIR. The residues involved in dimerization are shown in sticks and indicated.

its CNBD, responsible for their tetramerization. The last α-helix of the HCN channel superimposed well with the α2 helix of $Ps$PycTIR$_{CNBD}$, supporting their critical roles in oligomerization (Fig. 7e). $Ec$CRP, which functions as a dimer for transcriptional activation, has no additional α-helices at the N-terminus of CNBD unlike $Ps$PycTIR$_{CNBD}$ (Fig. 7f). Sequence alignment further demonstrated that most PycTIR family proteins have 21–30 amino acid N-terminal extensions compared with the CRP family proteins (Supplementary Fig. 9). Two highly conserved arginine residues, R4, and R12, were identified in the N-terminal extension of $Ps$PycTIR$_{CNBD}$ and form two hydrogen bonds with the main chain backbone and one salt-bridge with E114, respectively (Fig. 7g). To functionally validate the importance of the observed interactions, we created a single mutant $Ps$PycTIR$^{E114R}$, which will make electrostatic repulsion at $Ps$PycTIR oligomerization interface. Enzymatic analysis showed that in contrast to $Ps$PycTIR$^{WT}$, cUMP did not activate the NAD$^+$ cleavage activity of $Ps$PycTIR$^{E114R}$ (Fig. 6f). In phage infection experiments, oligomerization mutant $Ps$PycTIR$^{E114R}$ significantly attenuated the anti-phage ability of the $E.\ coli$ cells expressing corresponding $Ps$Pycsar system (Fig. 6h). In summary, these data demonstrated that disrupting the key ionic interaction (R12–E114)

between N-terminal CNBD domains of $Ps$PycTIR dimers severely interfered with the NADase activation of their C-terminal TIR domains and rendered the Pycsar system ineffective for phage defense, supporting the critical role of the N-terminal extension of PycTIR proteins for oligomerization and TIR activation. Despite these important findings, it should be noted that the TIR domain is absent from the complex structure of $Ps$PycTIR$_{CNBD}$-cUMP and thus the TIR-TIR interfaces that might also be important for filament formation cannot be analyzed and will need further investigation in the future.

## Discussion

In this study, we determined the crystal structures of a clade D PycC ($An$PycC) and a clade E PycC ($Ea$PycC) that synthesize cUMP and cCMP, respectively, and experimentally verified the important residues F60, K104, and F109 of $An$PycC, and F100, R142, and Q144 of $Ea$PycC for cUMP and cCMP production, respectively. Furthermore, we identified a zinc-binding ribbon at the C-terminus of $An$PycC, which is highly conserved among clade D PycC cyclases but are found to be absent in all the solved PycC structures, including $Pa$PycC (clade A), $Bc$PycC (clade B), and $Ea$PycC (clade E). Mutating three of the four zinc-binding

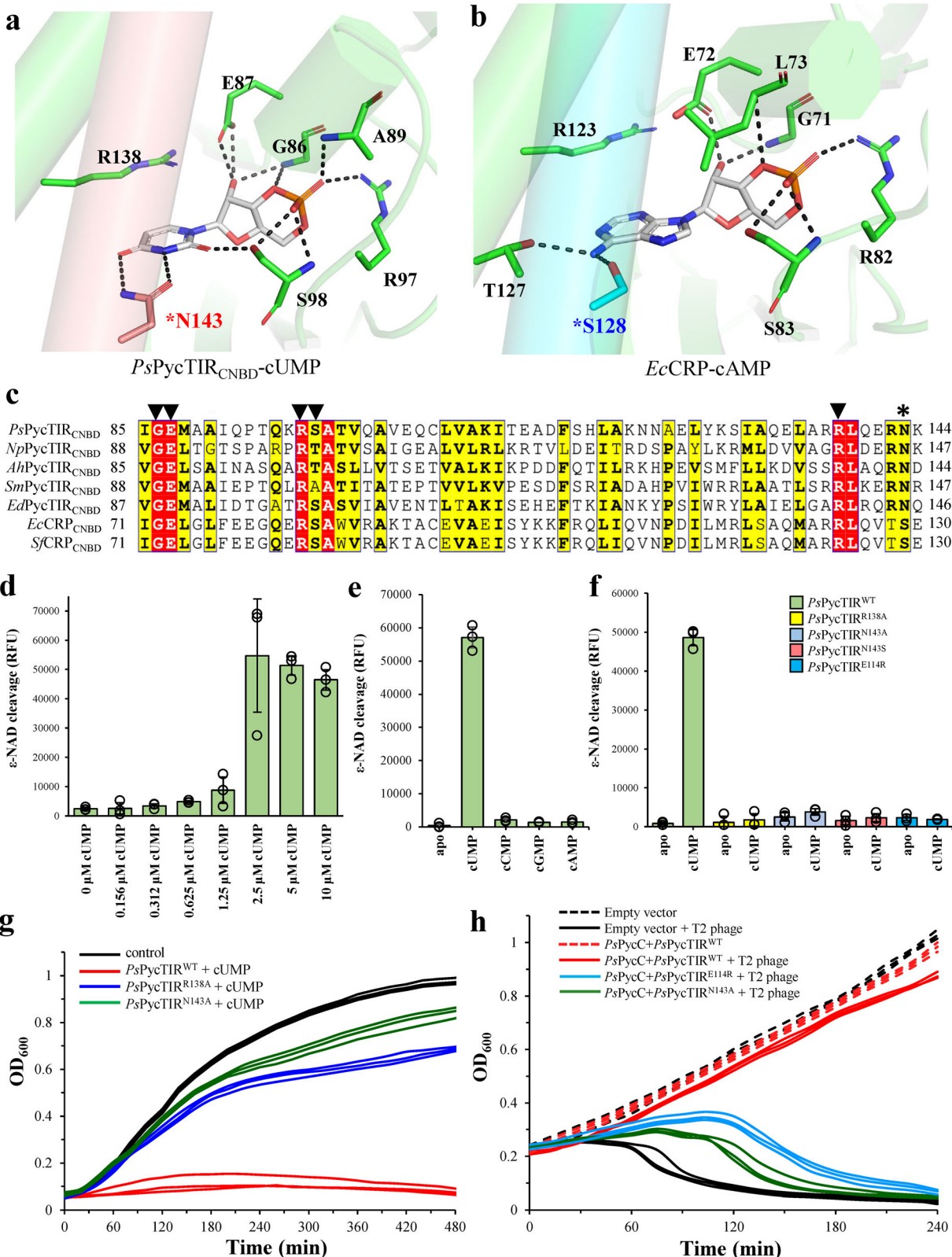

**a** *Ps*PycTIR$_{CNBD}$-cUMP **b** *Ec*CRP-cAMP

**c** Sequence alignment

**d**, **e**, **f** ε-NAD cleavage (RFU) bar charts

Legend (f): *Ps*PycTIR$^{WT}$, *Ps*PycTIR$^{R138A}$, *Ps*PycTIR$^{N143A}$, *Ps*PycTIR$^{N143S}$, *Ps*PycTIR$^{E114R}$

**g** OD$_{600}$ vs Time (min): control, *Ps*PycTIR$^{WT}$ + cUMP, *Ps*PycTIR$^{R138A}$ + cUMP, *Ps*PycTIR$^{N143A}$ + cUMP

**h** OD$_{600}$ vs Time (min): Empty vector; Empty vector + T2 phage; *Ps*PycC+*Ps*PycTIR$^{WT}$; *Ps*PycC+*Ps*PycTIR$^{WT}$ + T2 phage; *Ps*PycC+*Ps*PycTIR$^{E114R}$ + T2 phage; *Ps*PycC+*Ps*PycTIR$^{N143A}$ + T2 phage

residues abolished the production of cUMP by *An*PycC and render the *An*Pycsar-expressing *E. coli* cells sensitive to phage infections. Zinc finger has been known to stabilize a specific protein fold that function as interaction module to recognize DNA, RNA, or proteins[32]. As indicated by our in vitro enzyme activity assay, it is suggested that the zinc-binding ribbon of *An*PycC may stabilize certain structural element that is associated with production of cUMP.

Previous study reported that production of cCMP products by *E. coli* PycC is stimulated by phage infections. However, Tal et. al. also showed that *E. coli* PycC is auto-active in vitro and produce cCMP with pH range of 7.0–9.5. Similarly, we here demonstrated that purified *An*PycC proteins are active in vitro and generate cUMP using UTP at pH 9.0 but totally lose its enzyme activity at pH 6.5, suggesting the existence of active vs inactive conformation of *An*PycC (Supplementary

**Fig. 6 | Specific recognition of cUMP by PycTIR proteins. a, b** Enlarged view of the ligand-binding pocket of (**a**) cUMP-bound *Ps*PycTIR_CNBD and (**b**) cAMP-bound *Ec*CRP. The hydrogen-bonding interactions between them are shown in black dashed lines. The specificity-determining residues (N143 for uracil and S128 for adenine) are labeled with an asterisk (*). **c** Sequence alignment of CRP proteins with PycTIR proteins. The conserved residues interacting with cyclic mononucleotides are indicated by triangles (▼). The specificity-determining residue of PycTIR proteins and CRP proteins is indicated by an asterisk (*). **d** NAD$^+$ cleavage activity of *Ps*PycTIR under different cUMP concentrations (0, 0.15625, 0.3125, 0.625, 1.25, 2.5, 5, 10 μM). Source data are provided as a Source Data file. **e** NAD$^+$ cleavage activity of *Ps*PycTIR in the absence or presence of different cyclic mononucleotides at a concentration of 10 μM. Source data are provided as a Source Data file. **f** NAD$^+$ cleavage activity of wild-type and single mutant R138A, N143A, N143S and E114R of *Ps*PycTIR in the absence or presence of 10 μM cUMP. Source data are provided as a Source Data file. The data in (**d–f**) were shown as mean ± standard deviation for $n = 3$ independent replicates. **g** The growth curves of *E. coli* cells overexpressing *Ps*PycTIR$^{WT}$ (red), *E. coli* cells overexpressing R138A mutant of *Ps*PycTIR (blue), and *E. coli* cells expressing N143A mutant of *Ps*PycTIR (green). All of them were supplemented with 5 mM cUMP to medium. *E. coli* cells with empty vector serve as a negative control (black). Source data are provided as a Source Data file. **h** The growth curves of *E. coli* cells overexpressing wild-type *Ps*Pysar system containing *Ps*PycC and *Ps*PycTIR (red), *E. coli* cells overexpressing *Ps*Pysar system containing *Ps*PycC and *Ps*PycTIR$^{N143A}$ (green), *E. coli* cells overexpressing *Ps*Pysar system containing *Ps*PycC and *Ps*PycTIR$^{E114R}$ (cyan), and *E. coli* cells carrying empty vector (black) with T2 phage infection at an MOI of 0.01. The *E. coli* cells harboring *Ps*Pycsar system or empty vector without T2 phage infection serve as negative controls. Source data are provided as a Source Data file. The experiments in (**g**) and (**h**) were performed for $n = 3$ biological replicates and each of them is shown.

Fig. 3b–d). On the other hand, we presented a crystal structure of *An*PycC in the inactive state conformation in this study. Based on these data, we proposed that PycC cyclases are in a dynamic equilibrium between an inactive and an active conformation. In the absence of phage infections, the majority of PycC cyclases adopt inactive state conformation. Upon phage infection, PycC cyclases may directly bind certain phage components or indirectly sense the change of microenvironment within host cell, such as pH values, thereby triggering its conformational change from inactive to active conformation, making the catalytic centers accessible to pyrimidine nucleotides (Fig. 8a). The changed pH value probably promotes the formation or breakage of some ionic or hydrogen-bonding interactions within PycC cyclases, thereby triggering the conformational transition and stimulating the production of cUMP or cCMP products. In addition to the pH value and phage components, there may be other factors that can influence the cyclase activity of PycCs. It is reported that adenylate and guanylate cyclases isolated as monomers can undergo oligomerization to dimers to become active[33–35]. In this study, we found that purified *An*PycC proteins eluted as monomers in solution but form dimers in the crystals (Supplementary Fig. 10b). Whether oligomerization of PycC cyclases acts as another activation mechanism will need further studies and investigation in the future.

The Pycsar system has two types of downstream effectors: PycTM and PycTIR. This study provides the structural information regarding the bacterial PycTIR in both apo and cUMP-bound states. The binding of cUMP by PycTIR is probably an example of induce-fit: cUMP induces the conformational change of the helix and promotes the dimerization of PycTIR_CNBD, which subsequently strengthens the recognition of cUMP by inter-subunit contact, including the specificity-determining residue N143 (Fig. 8b). However, our ITC analysis revealed a 1:1 binding stoichiometry of cUMP and *Ps*PycTIR. Moreover, only one cUMP molecule is present in the crystal structure of the *Ps*PycTIR dimer. A previous study used nuclear magnetic resonance to show the sequential binding of two cAMP molecules to the *Ec*CRP dimer[36]. The authors proposed a "dynamics-driven allostery" model to explain the cooperative binding of cAMP, whereby the first cAMP activates the slow motion of *Ec*CRP, and the second cAMP completely quenches all protein motion[36]. Thus, the *Ps*PycTIR_CNBD-cUMP complex structure demonstrated here probably represent an intermediate structure wherein the first cUMP binds to and promotes the formation of dimeric *Ps*PycTIR_CNBD, unveiling the existence of a dynamic equilibrium between the ligand-free, monomeric *Ps*PycTIR_CNBD and the 2:2 cUMP-*Ps*PycTIR_CNBD complex (Fig. 8b). Furthermore, based on the gel-filtration data presented here, it is indicated that the cUMP-bound PycTIR dimers and tetramers work as basic repeating units, which could further extend into long filaments (Fig. 8c and Supplementary Fig. 10).

The cyclic nucleotide-binding domain (CNBD) is an ancient protein fold in all organisms for sensing the second messengers cAMP and cGMP[37–39]. CNBD, which can regulate numerous physiological

functions, is found in various types of receptor proteins, such as transcriptional regulator CRP, exchange protein activated by cAMP (Epac), cyclic nucleotide-gated channels, and HCN channels[40]. The serine or threonine residues that determine the specificity for cAMP recognition among CRP family proteins are well known[41]. In contrast, a CNBD-containing protein XC_0249, which specifically recognizes cGMP[42], has an alternative residue, E146, at the equivalent position for guanine base recognition compared with CRP (Fig. 6b and Supplementary Fig. 11). Herein, we provided key insights into cUMP recognition by a highly conserved asparagine residue in PycTIR proteins (Fig. 6). This mechanism for uracil base recognition via an asparagine residue was also discovered in a subclade of cGAS/DncV-like nucleotidyltransferase enzymes that specifically binds UTP to synthesize pyrimidine-containing cyclic dinucleotides[4].

## Methods
### Cloning, expression, and purification
The genes encoding full-length *Ea*PycC (NCBI accession: WP_049037095.1), *An*PycC (NCBI accession: WP_066377497.1), *An*PycTM (NCBI accession: WP_066377494.1), *Ps*PycC (GenBank: KZK76287.1) and *Ps*PycTIR (GenBank: KZK76288.1), and N-terminal CNBD truncation (a.a. 1–150) of *Ps*PycTIR, *Np*PycTIR (NCBI accession: WP_081473994.1), *Sm*PycTIR (GenBank: WP_080376336.1), and *Ah*PycTIR (GenBank: EPR88376.1) were *E. coli* codon optimized and chemically synthesized by Synbio Technologies Inc. (Supplementary Table 5). The mutants of *Ea*PycC, *An*PycC and *Ps*PycTIR were also synthesized by Synbio Technologies Inc (Supplementary Table 5). For crystallization, *Ea*PycC and *An*PycC was each cloned into pET32 vector to generate thioredoxin-tagged proteins, whereas *Ps*PycTIR_CNBD, *Np*PycTIR_CNBD, *Sm*PycTIR_CNBD, and *Ah*PycTIR_CNBD was each cloned into pET21 vector to generate C-terminal His₆-tagged proteins. For expression of *An*Pycsar system for in vivo experiments, *An*PycC and its variants was each cloned into pET24 vector, while *An*PycTM was cloned into pBAD vector. For expression of *Ps*PycTIR for *E. coli* toxicity assay, full-length *Ps*PycTIR or its variants was each cloned into pET24 vector. For expression of *Ps*Pycsar system for phage infection experiments, *Ps*PycC and *Ps*PycTIR (wild-type and its variants) were cloned into multiple cloning site 1 and 2 of pETDuet-1 vector, respectively. Cloning and plasmid construction were performed by Synbio Technologies Inc.

Recombinant proteins were produced using *E. coli* BL21 (DE3), which was cultivated in Luria Bertani (LB) broth at 37 °C. Protein overexpression was induced by addition of 0.5 mM IPTG until the OD_600 reached 0.6–0.8, followed by additional incubation for 16 hours at 16 °C. The cell pellets were harvested by centrifugation at 4 °C, 4500 x g for 30 min. To prepare C-terminal His₆-tagged PycTIR_CNBD proteins, a two-step purification procedure with Ni-affinity chromatography followed by size-exclusion chromatography (SEC) was applied. Briefly, the cell pellets were re-suspended in buffer A containing 50 mM Tris-HCl pH 8.0, 500 mM NaCl, 10 % glycerol, 1 mM tris(2-carboxyethyl)phosphine (TCEP) and 1 mM

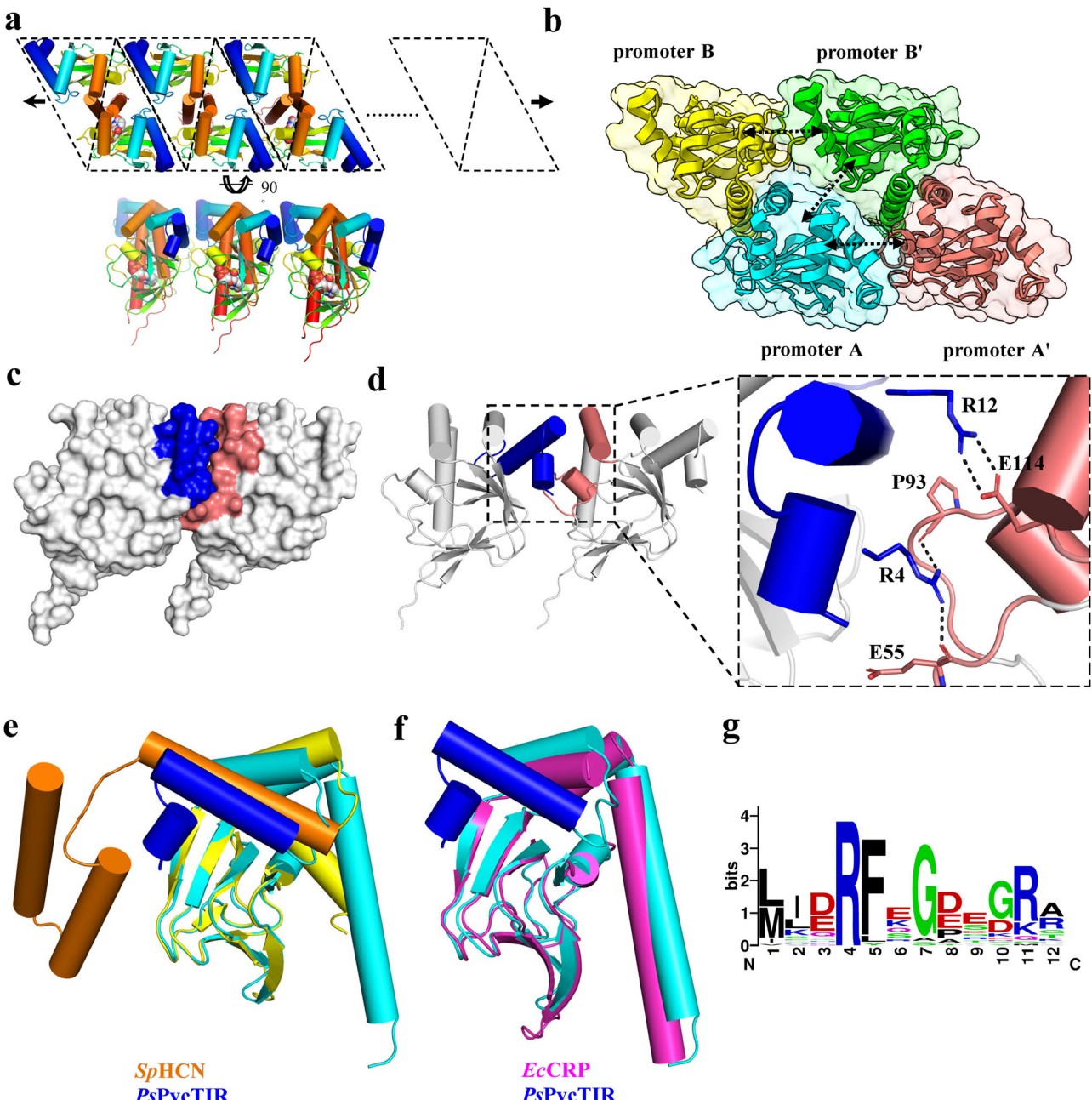

**Fig. 7 | Molecular mechanism underlying PycTIR oligomerization. a** Cartoon model of a *Ps*PycTIR linear filament implicated in the cUMP-bound *Ps*PycTIR$_{CNBD}$ crystal. **b** The dimer–dimer interactions of *Ps*PycTIR$_{CNBD}$. Each protomer is colored individually. The oligomerization interfaces between them are indicated by double arrows. **c, d** Surface (**c**) and cartoon (**d**) representation of protomer A and protomer A′ shown in (**b**). The regions involved in oligomerization are colored in blue (A) and salmon (A′), respectively. The enlarged view of the interface (A–A′) is shown in the right in (**d**). The interacting residues are shown in sticks and the salt-bridge (R12-E114) and two hydrogen bonds (R4-E55 and R4-P93) are shown in black dashed lines. **e** Superimposition of *Ps*PycTIR$_{CNBD}$ (cyan) with *Sp*HCN (yellow, PDB: 2PTM) with the N-terminal extensions colored in blue and orange, respectively. **f** Superimposition of *Ps*PycTIR$_{CNBD}$ (blue-cyan) with *Ec*CRP (magenta, PDB: 2CGP). **g** The conserved sequence logo of the N-terminal extension of PycTIR proteins.

phenylmethylsulfonyl fluoride (PMSF), lysed by sonication, and centrifuged at 4 °C for 30 min at the highest speed. The supernatant was filtered using 0.22 μM syringe filters (Millipore) and then loaded onto a 5 ml HisTrap HP column (Cytiva), followed by wash with buffer A plus 10 mM imidazole and subsequently elution using 20-200 mM imidazole gradient. Fractions containing the target protein were pooled and further purified by SEC using HiLoad Superdex 200 pg columns (Cytiva) equilibrated with gel buffer containing 20 mM Tris pH 8.0, 200 mM NaCl, 5% glycerol, 1 mM TCEP. To prepare full-length PycC proteins with an additional Gly residue at

N-terminal end, thioredoxin-fused proteins were first purified by Ni-affinity chromatography as described above, followed by Tobacco Etch Virus (TEV) protease cleavage at 4 °C for 2 to 3 days using reaction buffer containing 25 mM Tris 8.0, 100 mM NaCl, 1 mM DTT, 0.5 mM EDTA, 2% glycerol. After cleavage, PycC proteins were further separated from the thioredoxin-His$_6$-tag by SEC. Purified proteins were concentrated up to 20 mg/ml and store at −80 °C until use. SDS-PAGE analysis and size exclusion chromatography profiles for wildtype proteins and their variants are summarized in Supplementary Fig. 10.

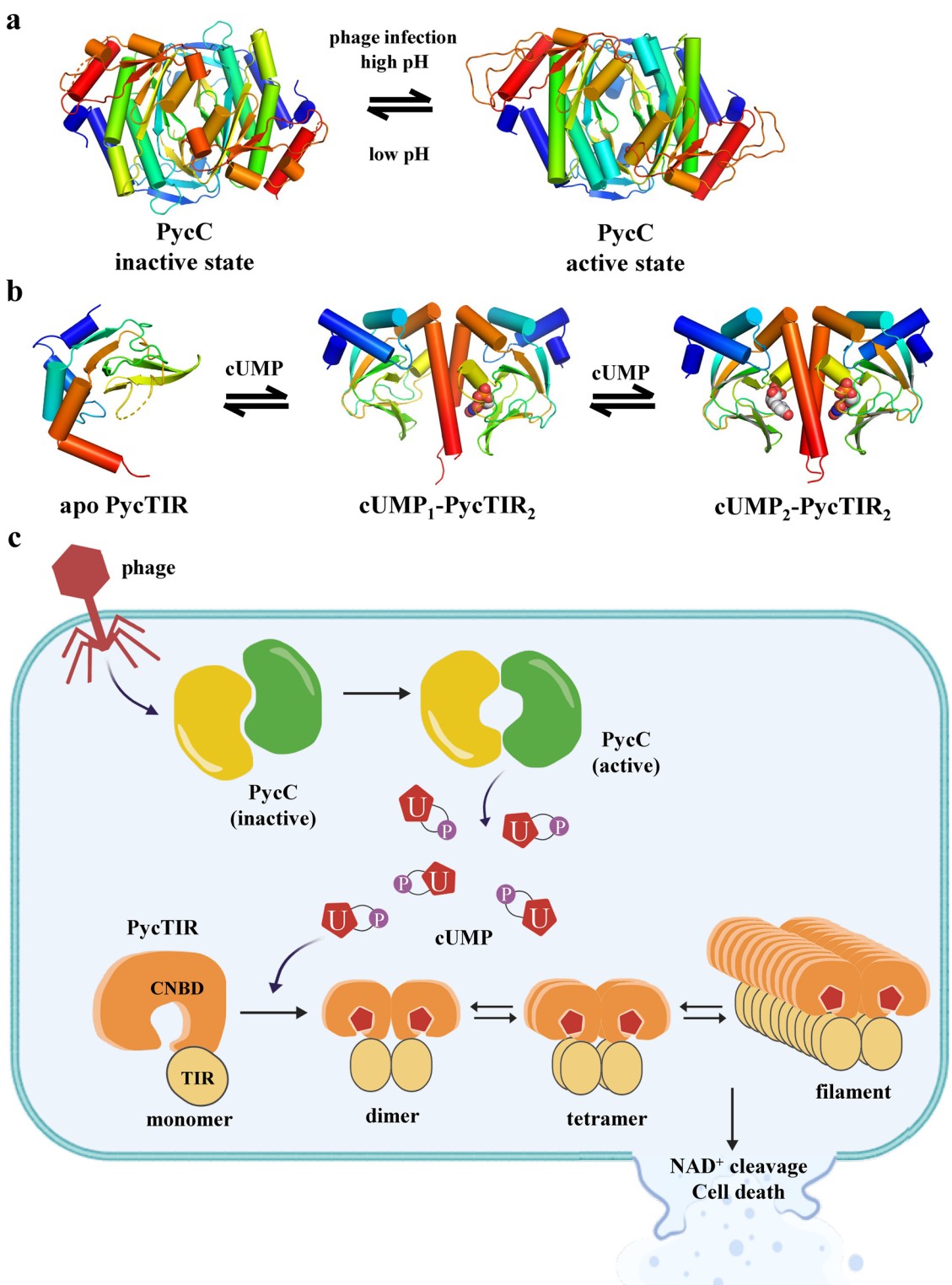

**Fig. 8 | Models for synthesis of cUMP by uridylate cyclases and NADase activation of effector PycTIR via cUMP binding and oligomerization. a** A proposed model explaining the conformational changes between active and inactive states of PycC cyclases followed by phage infections and/or pH changes. **b** A proposed dynamics-driven model describing the sequential binding of two molecules of cUMP (white spheres) into a dimeric PycTIR protein. **c** A schematic model describing the working mechanism of cUMP-activated Pycsar defense system.

## Crystallization and structure determination

The optimal protein concentration for crystallization was first determined using Pre-Crystallization Test (Cat No. HR2-141, Hampton). For crystallization of PycTIR_{CNBD}-cUMP complex, 2-fold molar excess of cUMP was added into the protein solution. Crystallization condition screening was conducted at 4 °C or 20 °C using sitting-drop vapor-diffusion method. The crystals of *Ea*PycC were grown in 0.05 M Bis-Tris pH 7.0, 1.6 M ammonium sulfate, 25% v/v glycerol, 0.15 M potassium chloride. The crystals of *An*PycC were grown in 0.1 M BIS-TRIS pH 6.5, 0.2 M Magnesium chloride hexahydrate, 25% w/v polyethylene

glycol 3,350 and were cryoprotected with reservoir solution supplemented with 15–20% glycerol. The crystals of *Np*PycTIR$_{CNBD}$ were grown in 0.2 M ammonium acetate, 0.01 M calcium chloride dihydrate, 0.05 M sodium cacodylate trihydrate pH 6.5, 10% w/v polyethylene glycol 4,000 and were cryoprotected with reservoir solution supplemented with 10–30% ethylene glycol. The complex crystals of *Ps*PycTIR$_{CNBD}$−cUMP were grown in 0.075 M HEPES pH 7.5, 7.5% w/v polyethylene glycol 8,000, 6% v/v ethylene glycol and 25% v/v glycerol. The X-ray diffraction data were collected and processed using HKL2000_v722[43] on beamline TPS 07 A of the National Synchrotron Radiation Research Center (NSRRC) in Hsinchu, Taiwan. The phase problem was solved by molecular replacement using *Bc*PycC structure (PDB: 7R65 for the two PycCs) or AlphaFold-generated models (for PycTIR) as searching template. The three-dimensional structures was iteratively refined and rebuilt using PHENIX 1.19.2-4158[44], *Coot* 0.9.6[45], and REFMAC 5.8.0267[46]. The collected data and refinement statistics are summarized in Supplementary Table 1–4 and the images of a portion of the electron density map for all the reported crystal structures are shown in Supplementary Fig. 12. All the figures containing three-dimensional structures of protein and ligand were depicted using either PyMOL 2.3.3[47] or UCSF ChimeraX 1.5[48].

### Isothermal titration calorimetry (ITC)
All the ITC experiments were conducted at 25 °C using nano-ITC (TA Instruments). Purified N-terminal CNBD truncation of PycTIR proteins were dialyzed extensively against buffer containing 50 mM Tris-HCl pH 8.0, 200 mM NaCl, 5% glycerol and 1 mM TCEP. 20 injections of cUMP or cCMP were sequentially titrated to PycTIR protein solution with a stirring speed of 300 rpm. The final data were analyzed using NanoAnalyze v3.12.5 (TA Instruments).

### NAD$^+$ cleavage activity assay
The NAD$^+$ cleavage activity of full-length *Ps*PycTIR and its variants were measured according to the previously described protocol[27]. Briefly, reaction mixture containing cyclic nucleotides (cUMP, cCMP, cGMP or cAMP) at the indicated concentration and 500 μM ε-NAD (No. N2630, Sigma-Aldrich) in the assay buffer (20 mM HEPES pH 7.5, 100 mM KCl) was pre-incubated at room temperature for 30 mins. Purified full-length *Ps*PycTIR proteins (wild-type, R138A, N143A, or N143S mutant) at a final concentration of 1 μM was then added to start the reaction. The fluorescence signal was continuously monitored for 1 hour with excitation/emission wavelengths of 300/410 nm. The changes of relative fluorescence units (RFUs) at the first 15 mins of the reaction were used for analysis.

### Protein toxicity analysis in *E. coli*
The *E. coli* cells harboring indicated expression plasmids were cultured overnight at 37 °C in LB broth, which were then diluted 1:50 in fresh LB broth to an O.D.$_{600}$ ranging from 0.1 to 0.2. Overexpression of *An*Pycsar system were induced by addition of 0.5 mM IPTG (for pET24) and 0.2% arabinose (for pBAD), while overexpression of *Ps*PycTIR alone was induced by 0.5 mM IPTG supplemented with 5 mM cyclic mononucleotides (cAMP, cGMP, cUMP, or cCMP) in the medium. The *E. coli* cultures were incubated for further 400–500 mins and the values of O.D.$_{600}$ was recorded every 20 mins. All the experiments were performed for $n = 3$ biological replicates.

### Phage infection assay
To access the anti-phage ability of *An*Pycsar and *Ps*Pycsar systems, *E. coli* cells harboring both pET24-*An*PycC and pBAD-*An*PycTM (*An*Pycsar system), *E. coli* cells harboring pETDuet-1-*Ps*PycC-*Ps*PycTIR (*Ps*Pycsar system) or control cells harboring empty vectors was each incubated at 37 °C with shaking overnight and then diluted with fresh medium containing 0.2% arabinose (for pBAD) and/or 0.5 mM IPTG (for pET24 and pETDuet-1). The *E. coli* cultures were continuously incubated at 37 °C until early log phase (O.D.$_{600}$ = 0.25) and infected with T2 phage at a multiplicity of infection (MOI) of 0.1 or 0.01. The values of O.D.$_{600}$ was recorded every 15 mins for 4 hours. All the experiments were performed for $n = 3$ biological replicates.

### Bioinformatic analysis
Multiple sequence alignment was performed using Clustal Omega[49]. To generate sequence logo shown in Figs. 2e, 3d, and 7g, 53 clade D PycC protein sequences and 31 PycTIR protein sequences were aligned using Clustal Omega, which then subjected to WebLogo (version 2.8.2)[50].

### LC-MS/MS analysis of in vitro reactions
To analyze the cyclic mononucleotide products synthesized by PycC cyclases, reaction mixtures containing 1 μM wild-type or mutant *An*PycC or *Ea*PycC with 0.5 mM indicated nucleotide substrates were prepared and incubated at 25 °C overnight. Reactions were heat inactivated at 90 °C for 10 mins, followed by centrifugation at 13000 × *g*, 4 °C for 30 mins. After removal of the pellet, 5 μl of the sample solution was injected into the LC-MS system for analysis.

A HPLC system (Ultimate 3000; Dionex, Germany), equipped with a pump (DGP 3600 M), an autosampler (WPS-3000T), and a BEH amide column (2.1 ×150 mm, 2.5 μm, Waters, Milford, MA, USA), was coupled with an electrospray ionization (ESI)-mass spectrometry (MS) (HCTultra PTM Discovery, Bruker Daltonics, Bremen, Germany) with a Compass 1.3 software (Bruker Daltonics, Bremen, Germany). The mobile phase A was composed of 10 mM ammonium bicarbonate in water and the mobile phase B was acetonitrile. A volume of 5 μL of sample was injected. After injection, the flow rate was set to 0.25 mL/min with 90% B in the mixture up to 0.5 min; from 0.5 to 7.0 min the percentage of mobile phase B was decreased from 90% to 40% and maintained at this percentage for 1 minute; then the percentage of eluent B increased to 90% in 0.2 min remaining in this condition for 3 min until the end of the run for column conditioning.

The ESI source was operated in the negative ion mode by setting capillary voltage at +4000 V, and End Plate Offset was set at −500V. Nitrogen was used as a nebulizing (40 psi) and drying gas (10 L/min, 350 °C). For the multiple-reaction monitoring (MRM) settings, the precursor ion was set as m/z 305.0 for cUMP and m/z 304.0 for cCMP with product ion mass range from m/z 50-700. Extraction ion chromatography (EIC) of the fragment ion was set as 111.0 for cUMP and 110.0 for cCMP. The data sets were post-processed using the Bruker Compass Data Analysis software (version 4.4, Bruker Daltonics, Bremen, Germany). All the LC-MS experiments were performed with two technical replicates ($n = 2$).

### Reporting summary
Further information on research design is available in the Nature Portfolio Reporting Summary linked to this article.

## Data availability
Data are available within the article and supplementary information. The coordinates and structure factors of *Ea*PycC, *An*PycC, *Np*PycTIR$_{CNBD}$, and *Ps*PycTIR$_{CNBD}$−cUMP complex generated in this study have been deposited in the Protein Data Bank under accession codes 8JSF, 8JSZ, 8JSJ, and 8JSK. The protein structures used for analysis in this study are available in the Protein Data Bank under accession codes 1WC1, 2CGP, 2PTM, 6YII, and 7R65. The LC-MS dataset generated in this study has been deposited in the MassIVE repository under accession code MSV000094839 [https://doi.org/10.25345/C50R9MF94]. Protein sequences used in this study are available in the NCBI database under accession code WP_049037095.1 (*Ea*PycC), WP_066377497.1 (*An*PycC), WP_066377494.1 (*An*PycTM), KZK76287.1 (*Ps*PycC), KZK76288.1 (*Ps*PycTIR), WP_081473994.1 (*Np*PycTIR), WP_080376336.1 (*Sm*PycTIR), and EPR88376.1 (*Ah*PycTIR). Source data are

provided as a Source Data file. Source data are provided with this paper.

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

## Acknowledgements
The authors thank National Synchrotron Radiation Research Center for beam time allocations and data collection supports. This work was supported by grants from the National Science and Technology Council (NSTC) in Taiwan [111-2311-B-039-001-MY3 to Y.C.].

## Author contributions
M.H.H. and C.J.C. designed and carried out the experiments and acquired and analyzed the data. C.S.Y. and Y.C.W. carried out the experiments. Y.C. interpreted the data, wrote the manuscript, and supervised the entire project.

## Competing interests
The authors declare no competing interests.
