## [Peer Review File · Nature Communications]

Structural and functional characterization of cyclic pyrimidine-regulated anti-phage systemREVIEWER COMMENTS

Reviewer #1 (Remarks to the Author):

Several prokaryotic immune defenses employ the use of nucleotide second messengers to protect against viral infection. Pycsar (pyrimidine cyclase system for antiphage resistance) is a recently described immune system that operates through specific production of the cyclic pyrimidine mononucleotides cyclic CMP and cyclic UMP generated by PycC cyclases. These unique signaling molecules are then selectively recognized by receptors that either cause membrane disruption (PycTM) or drive NAD⁺ degradation (PycTIR) ultimately leading to cell death or growth arrest resulting in protection against bacteriophages. In this manuscript, Yang et al. provide insight into the mechanisms at play which guide pyrimidine nucleotide substrate selectivity in the Pycsar cyclases and product recognition by the receptors. A series of structures determined using X-ray crystallography reveal the active site architecture of PycC homologs that had not previously been studied in-depth. Important new results include discovery of a zinc binding motif in clade D cyclases that provides structural stability and is also important for phage defense and the first experimentally determined structure of a putative cyclic CMP generating enzyme. Structures of the PycTIR effector cyclic nucleotide binding domain (CNBD) bound to cyclic UMP reveal a first look at the binding pocket features that are important for selectivity. Finally, a model for structural transitions that coincide with catalysis, receptor binding, and oligomerization is suggested.

The findings reported by Yang et al. are immediately relevant to the fields of cyclic nucleotide signaling, prokaryotic immunity, and structure biology and overall, this manuscript presents a clear story that serves to increase our understanding of an important component of prokaryotic immune defense. While initial enthusiasm is high, there are still several major concerns regarding the methodology and conclusions that if addressed would improve the quality of the manuscript and increase reader confidence in the author's interpretation of the data.

- Line 80- Why is uridine demoted to a parenthetical? The story is focused on the pyrimidine nucleotides so I would suggest a reframing of the introduction toward this end.
- The first results section introduces us to the structure of a PycC cyclase that putatively generates cyclic CMP. The authors have not shown explicitly that this enzyme makes cyclic CMP in the manuscript, nor has it been reported elsewhere in the literature. Indeed, this is also true for the AnPyC suspected of making cyclic UMP- it is not experimentally shown in the manuscript that it does make cyclic UMP but rather this activity is inferred from sequence alignment and clade designation.
- The different clades for Pycsar PycC cyclases need to be defined with at least a sentence or two and appropriate citation. The terms Clade E and D don't mean anything without context.
- The first sections of results: A structural analysis of PycC cyclases coupled with sequence alignment data was introduced first to the field in the original Pycsar discovery manuscript [Ref 23] (Tal & Morehouse et al., Cell 2021). It is unclear why the authors have here attempted to reintroduce the concept that Pycsar cyclases bear structural resemblance to other adenylate and guanylate

cyclases as this was already firmly established. Structural alignment of a putative cyclic CMP synthase, EaPycC, to other Pycsar cyclases is therefore more informative (PDB: 7R65 and 6YII) and so I would recommend a restructuring of the manuscript to move those structural superpositions from Supplementary Figures 3 and 4 to the main Figures 1 and 2.

- Lines 130-133: The additional N-terminal helix found on EaPycC is also a feature of BcPycC previously reported first in [Ref 23] (Tal & Morehouse et al., Cell 2021) and that should be acknowledged here and may warrant more in-depth comparison.
- Lines 140-143: The sentence lacks support. Mutations of these conserved residues could help provide that support.
- Line 155-157 & Figure 2E: No methods or description for how many sequences were used to generate the sequence logos.
- Line 158-160: Can the authors provide an explanation or speculation as to why in the absence of phage infection and in the absence of supplemental cyclic UMP, that overexpression of the *Anabaena* sp. Pycsar system would cause a cell growth defect?
- In the experiments regarding phage infection, the authors did not provide evidence or rationale for testing phage T2, and only T2. Were other phages also tested? Why was T2 selected and can the authors provide any quantitative assessment of the level of protection afforded by the selected Pycsar systems presented in this manuscript?
- There are issues with the phage infection experiments. An empty vector + phage control was not conducted for Figure 2G and 3F as was done in Figure 5H. This needs to be remedied so that there is a comparative “baseline” for infection without protection by Pycsar. Additionally, the traces for mutant + T2 infection condition in Figure 2G and 3F start at a lower OD600 than the other conditions (time = 0). The experiments need to be repeated such that all cultures start at the same initial OD600 in order for the results to be interpretable.
- While the methods section “Phage Infection Assay” does indicate n=3 biological replicates it is not indicated what the traces represent in the figures (average of 3? Individual trace representative of 3 biological replicates?). Additionally, were there technical replicates included in each experiment? It may be useful/necessary to indicate the spread (range) of the data shown in the growth curve figures.
- Line 178: The authors have made and tested a triple alanine mutant of the AnPycC cyclase. Were single point mutants also tested? Can the authors show that the triple mutant still expresses at a similar level to the wild type or that it is still properly folded?
- Lines 184-186: The strength of this sentence is severely limited because the authors have not shown that UTP is the preferred substrate for AnPycC nor that cUMP is actively produced during the T2 phage infections.
- Lines 270-271: What does excluding a surface area of approximately 1,100 Å² mean in this sentence? It is unclear which interfaces are being referenced and Figure 6a doesn't indicate these buried surfaces.
- Lines 298-299: There may be a typo here, but it is unclear what data is being referred to by this sentence. Is there a terminal residue that was mutated and tested?
- Lines 303-305: The importance of these residues is not validated as no direct binding was measured or observed.
- Lines 306-308: Cyclic UMP is not a confirmed product of AnPycC and therefore this sentence and the mechanisms of conformational change proposed in this manuscript are not supported by data.

The experimentally determined structure represents a possible “closed” state of the AnPycC cyclase but the “open” state is just a prediction.

- Line 321: Is the ratio expected to be 2:2 rather than 1:1 stoichiometry for cUMP if the CNBD resembles that of CRP and is likely a dimer once bound to cyclic nucleotide?
- Figure 1: the color scheme is not practical for the superpositions (too many similar overlapping colors)
- Figure 6: Panels C/D use salmon color for one protein and then in panel E of the same figure salmon is used for a different protein (SpHCN). This could be modified for clarity.
- The figure legends (and also the methods sections) describing the NADase assays do not indicate the amount of enzyme used. Additionally, the time point that is being analyzed is not indicated.
- Why was cyclic GMP not tested for NADase stimulation (Figure 5e)?
- It would be important to show some evidence that the CNBD mutants were still able to be expressed and purified to similar levels as wild type or else data regarding phage defense phenotypes could be inaccurate and actually reflect inactivity of a misfolded protein. Additionally, in support of the oligomerization mechanism proposed for PycTIRs, it would be important to show gels or size exclusion chromatography traces that NpPycTIR is a monomer and that PcPycTIR is a dimer or that there is some dynamic equilibrium that is influenced by cyclic UMP binding.
- Inconsistent use of the abbreviation “LB” – Line 369/370 defined LB as “Luria Bertani broth” and Line 428 defined LB as “lysogeny broth broth”
- Line 400: Need to use specifics when suggesting crystals were cryoprotected. Which crystals were cryoprotected with 15-20% glycerol and which with 10-30% ethylene glycol? The crystallization condition for EaPycC already had 25% glycerol and for PsPycTIR-CNBD-cUMP already had 6% ethylene glycol and 25% glycerol.

Reviewer #2 (Remarks to the Author):

Pycsar (pyrimidine cyclase system for anti-phage resistance) is an antiviral system in bacteria in which a PycC enzyme is activated during phage infection to synthesize a cUMP (3',5'-cyclic uridine monophosphate) or cCMP (3',5'-cyclic cytidine monophosphate) second messenger. A Pycsar effector protein encoded in the same operon recognizes this second messenger through a CNBD (cyclic nucleotide binding domain) and executes abortive infection through either a TIR (Toll/interleukin-1 receptor) NADase domain or membrane-disrupting TM (transmembrane) domains. The mechanism by which PycC enzymes sense phage infection is currently unknown. Furthermore, the molecular details of how Pycsar effectors discriminate between the cNMPs produced by cognate PycC cyclases and “house-keeping” messengers such as cGMP and cAMP remain poorly characterized.

Here, Wang, Yang et al. present crystal structures of the clade E cyclase EaPycC and the clade D cyclase AnPycC. In their structure of AnPycC they identify a zinc-finger motif common to most clade D PycC enzymes and claim that this structural element is required for AnPycC activity during

phage infection. Wang, Yang et al. further determine the structure of two CNBDs from PycTIR effector proteins: apo NpPycTIRC�BD and PsPycTIRC�BD in complex with cUMP. Using ITC analysis they demonstrate that several PycTIR proteins recognize cUMP with high affinity. Through their structure of cUMP-bound PsPycTIRC�BD and structure-guided sequence alignment the authors identify the structural determinants of cUMP specificity in CNBDs of PycTIR effectors. Wang, Yang et al. further use their crystal structure of cUMP-bound PsPycTIRC�BD to infer the mechanism of PsPycTIR filament formation upon cUMP recognition.

The structures and analysis presented here do not provide many new insights to the field of Pycsar immunity or bacterial anti-phage defense. Crystal structures of Clade A PaPycC (Linder et al. J. Struc. Biol 2020) and Clade B BcPycC (Tal, Morehouse et al. Cell 2021) have previously been reported. Structural, biochemical, and bioinformatic analysis by Tal, Morehouse et al. previously characterized the conserved active site residues of PycC enzymes and the biochemical requirements for cNMP production. One novel element of this manuscript is the structure and analysis of PsPycTIR bound to cUMP, which provides key insights into specific second messenger recognition by this important family of bacterial immune effectors. Overall, this manuscript requires revisions, including the addition of key biochemical experiments, to be suitable for publication in Nature Communications.

Major Points of Review

1. The authors identify key features in the EaPycC and AnPycC crystal structures, including residues presumably involved in UTP and CTP recognition (previously reported) and unique structural features including an N-terminal $\alpha 1$ helix (previously reported), a “specific Y-shaped structure comprising two anti-parallel β -sheets ($\beta 10$ – $\beta 13$)” in EaPyc and a zinc-ribbon motif in AnPycC.
 - Crystal structures of PaPycC (Linder et al. J. Struc. Biol 2020) and BcPycC (Tal, Morehouse et al. Cell 2021) were previously determined. It would significantly improve the clarity of the manuscript and better contextualize the manuscripts findings for the authors to compare (in the “Results” text and the figures) their EaPycC and AnPycC structures to PaPycC and BcPycC rather than more distantly related proteins.
 - The authors must perform biochemical experiments to analyze the production of cCMP and cUMP by EaPycC and AnPycC respectively. Previous work in the field (Tal, Morehouse et al.) demonstrates that most PycC enzymes are auto-active in vitro.
 - o For all key structures features the authors identify (residues in the catalytic site/ nucleotide binding pocket, the “Y-shaped structure” in EaPycC, the zinc-ribbon in AnPycC) mutations must be made and tested for their effect on cNMP synthesis.
 - o For the authors to claim a role for the zinc-ribbon in AnPycC activity during phage infection, they must investigate its role in cUMP synthesis. If this cannot be done through in vitro experiments the authors can use mass-spectrometry to detect cUMP in the lysates of E. coli expressing WT or mutated AnPycC during phage infection.
 - Crystal structures of EaPycC or AnPycC in complex with CTP/ UTP substrates would add novelty to the manuscript.

2. The authors measure the growth of E. coli cells expressing an empty vector or Pycsar systems

(WT and mutated) during T2 phage infection (Figs 2g, 3f, 5h).

- These experiments are missing critical controls.

o The authors must show the growth curves of empty vector cells infected with T2 phage (Figs 2g,3f), as in Fig. 5h.

o The authors should also show the growth curves of Pycsar-expressing E. coli in the absence of phage infection (Figs 2g,3f,5h)

- The authors should perform an outgrowth of their overnight cultures prior to growth curve measurements in order to ensure each culture is in early log phase at the beginning of the experiment.

- The authors must normalize the density of all cultures at the beginning of growth curve measurements. In both Fig. 2g and Fig. 3f the cultures start at different OD600 measurements at timepoint 0 and thus it is not possible to compare the growth of different cultures over time.

3. The authors reveal the PsPycTIRC�BD dimerization interface in their crystal PsPycTIRC�BD – cUMP crystal structure. Based on the crystal packing of PsPycTIRC�BD, they further propose that contacts between A' and A, A and B', and B' and B protomers in adjacent dimers mediate PsPycTIR linear filament formation.

- It was previously shown by electron microscopy that PycTIR proteins form filaments specifically in the presence of their cognate cNMP ligand (Tal, Morehouse, et al., Cell 2021).

- The crystal structure contains only the CNBD and not the TIR domain of PsPycTIR, and thus it is not clear that this structure provides relevant insight into filament formation of the full-length protein. Previous structures of nucleotide-bound SAVED-TIR and STING-TIR proteins reveal important TIR-TIR contacts involved in filament formation (Hogrel et al., Nature 2022; Morehouse et al., Nature 2022).

- To demonstrate the relevance of the proposed A' and A, A and B', and B' and B interfaces for filament formation, the authors should analyze WT vs. interface mutant (e.g. E114R) PsPycTIR oligomerization using an in vitro assay such as SEC-MALS.

Minor Points of Review

1. The authors speculate that “the ligand-free AnPycC presented here is in its inactive state with closed catalytic centers and may be activated by the binding of phage proteins” (lines 171-173). However the authors’ data also demonstrates that when heterologously expressed in E. coli, AnPycC WT and its effector AnPycTM are extremely toxic (Fig. 2f, 3e), suggesting auto-activity by the WT AnPyc system in the absence of phage proteins.

- To resolve whether AnPycC is active in the absence of phage proteins, the authors should test the activity of the purified AnPycC protein in vitro (see major point 1).

- It is intriguing that the AlphaFold model of AnPycC is in an “open”, auto-active form in contrast to the “closed” crystal structure of AnPycC. Is it possible that one of the co-purifying molecules in the AnPycC crystal is inducing the “closed” conformation, potentially by mimicking an inhibitory ligand?

2. It might be helpful if the authors made a graph to summarize the KD of different PycTIR proteins for Fig. 4a, as measured by ITC (Supp Fig. 6), in place of the ITC analysis for only PsPycTIR that is currently shown in Fig. 4a.

- It would be interesting, but not essential, for the authors to test the affinity of PycTIR proteins for other cyclic mononucleotides in comparison to cUMP to demonstrate specificity of these effectors for their cognate second messenger.

3. The most novel element of this manuscript is the identification of residues which control specific recognition of cNMP messengers by PycTIR effectors. The authors could provide further experiments related to this finding to enhance their manuscript.

- The authors could use ITC to analyze the affinity of PsPycTIR cUMP-binding mutants for cUMP and other cNMPs. The authors could make analogous mutations in the other PycTIR proteins tested in this manuscript to demonstrate the conserved importance of these residues for cNMP recognition.

4. T2 phage infection of cells expressing a PsPycTIR cUMP-recognition mutant and putative oligomerization mutant, which should be unable to be activated for NADase activity, do not appear significantly more susceptible to phage infection (Fig. 5h). The authors should address this.

5. Text comments

- The authors should include the residue numbers for each protein in the sequence alignment in Fig. 5c.

- Line 223: “invisible” should be written as “unresolved in the NpPycTIRC�BD crystal structure”.

- Line 669: “regulate” should be corrected to “regulates”.

- Line 798: “yelloworange” should be corrected to “yellow-orange”.

- The authors should reference the following paper for use of AlphaFold structural predictions:
Jumper et al. Nature 2021

Author Rebuttals

Reviewer #1 (Remarks to the Author):

Several prokaryotic immune defenses employ the use of nucleotide second messengers to protect against viral infection. Pycsar (pyrimidine cyclase system for antiphage resistance) is a recently described immune system that operates through specific production of the cyclic pyrimidine mononucleotides cyclic CMP and cyclic UMP generated by PycC cyclases. These unique signaling molecules are then selectively recognized by receptors that either cause membrane disruption (PycTM) or drive NAD⁺ degradation (PycTIR) ultimately leading to cell death or growth arrest resulting in protection against bacteriophages. In this manuscript, Yang et al. provide insight into the mechanisms at play which guide pyrimidine nucleotide substrate selectivity in the Pycsar cyclases and product recognition by the receptors. A series of structures determined using X-ray crystallography reveal the active site architecture of PycC homologs that had not previously been studied in-depth. Important new results include discovery of a zinc binding motif in clade D cyclases that provides structural stability and is also important for phage defense and the first experimentally determined structure of a putative cyclic CMP generating enzyme. Structures of the PycTIR effector cyclic nucleotide binding domain (CNBD) bound to cyclic UMP reveal a first look at the binding pocket features that are important for selectivity. Finally, a model for structural transitions that coincide with catalysis, receptor binding, and oligomerization is suggested.

The findings reported by Yang et al. are immediately relevant to the fields of cyclic nucleotide signaling, prokaryotic immunity, and structure biology and overall, this manuscript presents a clear story that serves to increase our understanding of an important component of prokaryotic immune defense. While initial enthusiasm is high, there are still several major concerns regarding the methodology and conclusions that if addressed would improve the quality of the manuscript and increase reader confidence in the author's interpretation of the data.

• Line 80- Why is uridine demoted to a parenthetical? The story is focused on the pyrimidine nucleotides so I would suggest a reframing of the introduction toward this end.

Response: Thanks for reviewer's comments. The introduction has been rewritten to focus on the pyrimidine nucleotides. The modifications are described as below:

1. To focus on the pyrimidine nucleotides, the original first paragraph in the introduction section describing the function of cAMP and cGMP in prokaryotes and eukaryotes have been removed. Instead, we added a general description in lines 80–87 and shown as below:

“Billions of years of coevolution between bacteria and phages have led to the development of more than 100 bacterial defense systems against phage infections¹. Meanwhile, phages have evolved proteins to counteract bacterial defenses, thus allowing phage survival. As a consequence,

nucleotide signal-mediated bacterial defense systems evolved, which are highly diverse in their responses, giving bacteria survival advantage². Purine nucleotide signals, such as 3',5'-cyclic adenosine monophosphate (cAMP) and 3',5'-cyclic guanosine monophosphate (cGMP), have been subject to research for several decades. In contrast, pyrimidine-containing nucleotide signals, including cyclic mono-pyrimidines³ and cyclic di-pyrimidines^{4,5}, have only recently been discovered and appreciated.”

2. To give a clear picture of how cUMP/cCMP-mediated Pycsar defense system works in bacteria and the bottleneck of current studies, we have added descriptions in the introduction section, lines 108–129 and shown as below:

“Upon phage infections, produced cyclic mono-pyrimidines activates their receptors PycTM, which disrupts membrane integrity by forming transmembrane-spanning pore, or PycTIR, which deplete NAD⁺, leading to growth arrests and cell deaths, thus protecting the uninfected bacterial populations, a strategy called “abortive infection”¹⁹. Pyrimidine cyclase PycCs share conserved catalytic motifs with class III adenylate and guanylate cyclase enzymes and require divalent cations, such as Mg²⁺ or Mn²⁺ for catalysis²⁰. Based on phylogenetic analysis, pyrimidine cyclase PycCs were assigned to five subclades from clade A to clade E. Clade A PycC cyclases have two consecutive nucleotide cyclase domains, whereas clade B–E cyclases contains only one nucleotide cyclase domain³. However, due to the limited structural information of PycC cyclases, the molecular mechanisms underlying the synthesis of cUMP and cCMP remain unclear. In Pycsar defense system, the receptor PycTIR is composed of a N-terminal cyclic nucleotide-binding domain (CNBD) and a C-terminal TIR domain. Binding of cUMP to its N-terminal CNBD promotes the oligomerization of PycTIR proteins, followed by the activation of its NADase activity of the C-terminal TIR domain. Depletion of the essential cofactor NAD⁺ by PycTIR proteins will lead to growth arrests and even cell death of the bacterial hosts, stopping the phage replication and propagation. Previous study has shown that the NAD⁺ cleavage activity of PycTIR proteins could be stimulated by little amount of cUMP (50–100 nM); in contrast, other cyclic mononucleotides, even at concentrations up to 1 mM, still cannot activate PycTIR proteins³. Furthermore, *in vivo* phage challenge assays demonstrated that ectopic addition of cUMP activated PycTIR-expressing *E. coli* cells for phage defense, whereas the addition of cCMP did not. All these results showed the specificity of the CNBD of PycTIR to cUMP³. However, the detailed molecular mechanism underlying specific cUMP recognition and oligomerization by PycTIR proteins remains poorly understood.”

• *The first results section introduces us to the structure of a PycC cyclase that putatively generates cyclic CMP. The authors have not shown explicitly that this enzyme makes cyclic CMP in the manuscript, nor has it been reported elsewhere in the literature. Indeed, this is also true for the AnPyC suspected of making cyclic UMP- it is not experimentally shown in the manuscript that it does make cyclic UMP but rather this activity is inferred from sequence alignment and clade designation.*

Response: Thanks for reviewer's comments. To experimentally characterize the production of cyclic pyrimidine products by *EaPycC* and *AnPycC*, we further performed *in vitro* enzymatic reactions by incubating purified *EaPycC* proteins with CTP and purified *AnPycC* proteins with UTP. LC-MS/MS analysis identified the products, cyclic CMP and cyclic UMP in the reaction mixture containing *EaPycC* and *AnPycC*, respectively. The results have been newly added in Supplementary Fig. 1 and 3 and shown as below.

Supplementary Figure 1. Extraction ion chromatography (m/z 304.0 \rightarrow m/z 109.9) and MS/MS spectrum of the LC-MS/MS analysis of (a) the chemical standard of cCMP and (b) cCMP produced in the overnight reaction containing 1 mg/ml *EaPycC* and 1 mM CTP at pH 9.0.

Supplementary Figure 3. Extraction ion chromatography (m/z 305.0 \rightarrow m/z 110.8) and MS/MS spectrum of the LC-MS/MS analysis of (a) the chemical standard of cUMP and (b) cUMP produced in the overnight

reaction containing 1 mg/ml *AnPycC* and 1 mM UTP at pH 9.0.

• *The different clades for Pycsar PycC cyclases need to be defined with at least a sentence or two and appropriate citation. The terms Clade E and D don't mean anything without context.*

Response: Thanks for reviewer's comments. The description regarding the different clades for PycC cyclases in Pycsar defense systems have been newly added in the introduction section, lines 113–116, cited with reference 3 (Tal & Morehouse et al., Cell 2021), and shown as below:

“Based on phylogenetic analysis, pyrimidine cyclase PycCs were assigned to five subclasses from clade A to clade E. Clade A PycC cyclases have two consecutive nucleotide cyclase domains, whereas clade B–E cyclases contains only one nucleotide cyclase domain³”.

• *The first sections of results: A structural analysis of PycC cyclases coupled with sequence alignment data was introduced first to the field in the original Pycsar discovery manuscript [Ref 23] (Tal & Morehouse et al., Cell 2021). It is unclear why the authors have here attempted to reintroduce the concept that Pycsar cyclases bear structural resemblance to other adenylate and guanylate cyclases as this was already firmly established. Structural alignment of a putative cyclic CMP synthase, *EaPycC*, to other Pycsar cyclases is therefore more informative (PDB: 7R65 and 6YII) and so I would recommend a restructuring of the manuscript to move those structural superpositions from Supplementary Figures 3 and 4 to the main Figures 1 and 2.*

Response: We greatly thanks for reviewer's suggestions. To give more insights into how PycC cyclases works, the first sections of results have been rewritten. The descriptions regarding structural comparison of the cyclic CMP synthase *EaPycC* with other Pycsar cyclases, including *BcPycC* (PDB: 7R65) and *PaPycC* (PDB: 6YII), have been newly added in the results sections, lines 150–164, and described as below:

“Structural comparative analysis showed that *EaPycC* share similar nucleotide cyclase core domain with both *PaPycC* and *BcPycC* with root-mean-square deviations (RMSDs) of 3.7 Å and 2.7 Å for 205 and 186 matched C α pairs, respectively (Supplementary Fig. 2a, b). The N-terminal α helix ($\alpha 7'$) previously identified to be a hallmark of PycC cyclases, is also present in *EaPycC* (Fig. 1c–e). However, unlike *BcPycC*, the N-terminal α helix ($\alpha 1$) of *EaPycC* packs against $\alpha 4$ and $\alpha 5$ helices in the same protomer instead of extending into the opposing protomer (Fig. 1c). Another unique feature found in *EaPycC* is a C-terminal Y-shaped structure that comprises two anti-parallel β -sheets ($\beta 10$ – $\beta 13$) near the active sites (Fig. 1c, d). In contrast, *BcPycC* only has a simpler β -hairpin structure at the equivalent position (Fig. 1c, d). Similarly, the unique C-terminal Y-shaped structure of *EaPycC* is also absent in *PaPycC* structure (Fig. 1e). The active site of *EaPycC* contains two conserved catalytic residues D102 and D146 responsible for divalent metal coordination²⁰, which superimposed well with those in *PaPycC* and *BcPycC* (Supplementary Fig. 2c). In contrast to the UTP-recognition residues, Y50, D94, and R97,

identified in *BcPycC*, *EaPycC* contains strikingly different specificity-determining residues, F100, R142, and Q144, for CTP binding (Supplementary Fig. 2d)”. These results have been newly added in Figure 1c–1e and Supplementary Fig. 2 and shown as below.

As the reviewer’s suggestion, structural superpositions of *AnPycC* with *BcPycC* and *PaPycC* from original Supplementary Figures 3 and 4 have been combined and moved to the revised Figure 4 and shown as below.

Figure 1. Structural characterization of a cCMP-synthesizing PycC cyclase, *EaPycC*.

(c) The front (left) and back (right) view of the superimposed *EaPycC* and *BcPycC* promoters. The missing residues (a.a. 22–66) of *EaPycC* protomer are indicated by black dashed lines. The inlet shows the enlarged view of the superimposed Y-shaped structure of *EaPycC* with β -hairpin of *BcPycC* at the equivalent position. (d, e) Structural alignment of *EaPycC* dimer with (d) *BcPycC* dimer, and (e) *PaPycC* dimer. *EaPycC*, *BcPycC*, and *PaPycC* proteins are colored in salmon, cyan, and pale-green, with their unique N- and C-terminal structural elements colored in red, blue, and forest-green respectively.

Supplementary Figure 2. Structural comparison of *EaPycC* (salmon) with *PaPycC* (pale green, PDB 6YII) and *BcPycC* (cyan, PDB 7R65).

Ribbon representation of the superimposed *EaPycC* with (a) *PaPycC* and (b) *BcPycC* using DALI server. For clarity, only the nucleotide cyclase core of them is shown. (c) Superimposed catalytic aspartate residues of *EaPycC* with those in *PaPycC*, *BcPycC* and *AnPycC* model (yellow-orange). The calcium and manganese ion in the *PaPycC* structure are shown in grey and magenta sphere, respectively. (d) Superimposed active-site residues of both *EaPycC* and *BcPycC*.

Figure 4. Structural comparison of *AnPycC* with other PycC pyrimidine cyclases.

(a) Structural comparison of *AnPycC* protomer with *BcPycC* protomer. (b) Structural comparison of *AnPycC* dimer with *BcPycC* dimer. (c) Left, superimposed *AnPycC* dimer with *PaPycC*. Right, the enlarged view of the superimposed C-terminal structures of *AnPycC* and *PaPycC*. (d) Left, superimposed *AnPycC* dimer with *EaPycC* dimer. Right, the enlarged view of the superimposed C-terminal structures of *AnPycC* and *EaPycC*. *AnPycC*, *BcPycC*, *PaPycC*, and *EaPycC* proteins are colored in pale-green, sky-blue, salmon, and yellow with their unique N- and C-terminal structural elements colored in forest-green, blue, red, and orange respectively. The zinc-coordinating residues of *AnPycC* located at $\alpha 9$ helix and $\beta 10$ - $\alpha 7$ loop are shown in sticks and the bound zinc ions are shown in grey spheres.

• Lines 130-133: The additional N-terminal helix found on *EaPycC* is also a feature of *BcPycC* previously reported first in [Ref 23] (Tal & Morehouse et al., Cell 2021) and that should be

acknowledged here and may warrant more in-depth comparison.

Response: Thanks for reviewer's comments. Structural comparison of *EaPycC* with *BcPycC* has been newly added in Figure 1c–d and the descriptions regarding the additional N-terminal helix found on both *EaPycC* and *BcPycC* has been added in lines 153–156, and shown as below:

“The N-terminal α helix ($\alpha 7'$) previously identified to be a hallmark of PycC cyclases, is also present in *EaPycC* (Fig. 1c–e). However, unlike *BcPycC*, the N-terminal α helix ($\alpha 1$) of *EaPycC* packs against $\alpha 4$ and $\alpha 5$ helices in the same protomer instead of extending into the opposing protomer (Fig. 1c).”

• *Lines 140-143: The sentence lacks support. Mutations of these conserved residues could help provide that support.*

Response: Thanks for reviewer's comments. To support the crystallographic observations, we made a triple alanine mutant of *EaPycC* (F100A/R142A/Q144A) and tested whether this mutant affect cCMP production. The following figure shows that no cCMP products were detected in the reaction mixture containing *EaPycC*^{F100A/R142A/Q144A} mutant and CTP substrates, validating the critical roles of these residues for CTP recognition.

Figure. Extraction ion chromatography of cCMP signals (m/z 304.0→ m/z 109.9) in the LC-MS/MS result of the overnight reaction containing 1 mg/ml *EaPycC*^{F100A/R142A/Q144A} and 1 mM CTP at pH 9.0. The cCMP signal is not observed in the sample.

• *Line 155-157 & Figure 2E: No methods or description for how many sequences were used to generate the sequence logos.*

Response: The descriptions for generation of the sequence logos shown in Fig. 2e, Fig. 3d, and Fig. 7g were added in method section, line 485–487 of revised manuscript, and shown as below:

“To generate sequence logo shown in Fig. 2e, Fig. 3d, and Fig. 7g, 53 clade D PycC protein sequences and 31 PycTIR protein sequences were aligned using Clustal Omega, which then subjected to WebLogo (version 2.8.2)”.

• *Line 158-160: Can the authors provide an explanation or speculation as to why in the absence of*

phage infection and in the absence of supplemental cyclic UMP, that overexpression of the Anabaena sp. Pycsar system would cause a cell growth defect?

Response: Thanks for reviewer's comment. As mentioned above, we have further demonstrated that purified *AnPycC* proteins are active *in vitro* and generate cUMP using UTP at pH 9.0 in the absence of any phage components, suggesting the existence of active conformation of *AnPycC* proteins. On the other hand, we presented a crystal structure of *AnPycC* protein in the inactive conformation in this manuscript. Based on these data, it is proposed that *AnPycC* proteins exist in two conformational states: one in inactive conformation and the other in active conformation. Accordingly, it is suggested that the recombinantly expressed *AnPycC* proteins inside *E. coli* cells is in a dynamic equilibrium between these two states and the proportion of active *AnPycC* proteins probably produce enough amounts of cUMP to activate the co-expressed effector *AnPycTM*, which cause a cell growth defect.

• In the experiments regarding phage infection, the authors did not provide evidence or rationale for testing phage T2, and only T2. Were other phages also tested?

Response: Thanks for reviewer's comments. Due to limited manpower and resources in our lab, we have initially selected two to three different phages to test their protective effects. We found that both *AnPycsar* and *PsPycsar* defense systems provided the highest defense against T2 phage. Therefore, T2 phage was chosen for subsequent phage infection experiments presented in our manuscript.

• There are issues with the phage infection experiments. An empty vector + phage control was not conducted for Figure 2G and 3F as was done in Figure 5H. This needs to be remedied so that there is a comparative "baseline" for infection without protection by Pycsar. Additionally, the traces for mutant + T2 infection condition in Figure 2G and 3F start at a lower OD600 than the other conditions (time = 0). The experiments need to be repeated such that all cultures start at the same initial OD600 in order for the results to be interpretable.

Response: Thanks for reviewer's comments. The control experiments for growth curves of *E. coli* cells carrying empty vector with T2 phage infection have been conducted for Figure 2g and 3f. To compare the growth of different cultures over time, we have repeated the phage infection experiments such that all cultures start at nearly the same initial OD600 in order for the results to be interpretable ($\Delta OD_{600} < 0.03$). The new results have been updated in revised Figure 2g and 3f and shown as below.

Figure 2g. The growth curves of *E. coli* cells overexpressing wild-type *AnPycC* and *AnPycTM* (orange), *E. coli* cells overexpressing *AnPycC* triple mutant (C221A/C223A/C260A) and *AnPycTM* (purple), and *E. coli* cells harboring empty vector (black) with T2 phage infection at an MOI of 0.1. The *E. coli* cells harboring *AnPycsar* system or empty vector without T2 phage infection and protein induction serve as negative controls.

Figure 3f. The growth curves of *E. coli* cells overexpressing wild-type *AnPycC* and *AnPycTM* (orange), *E. coli* cells overexpressing *AnPycC* triple mutant (F60A/K104A/F109A) and *AnPycTM* (green), and control cells (black) with T2 phage infection at an MOI of 0.1. The *E. coli* cells harboring *AnPycsar* system or empty vector without T2 phage infection and protein induction serve as negative controls.

• While the methods section “Phage Infection Assay” does indicate $n=3$ biological replicates it is not indicated what the traces represent in the figures (average of 3? Individual trace representative of 3 biological replicates?). Additionally, were there technical replicates included in each experiment? It may be useful/necessary to indicate the spread (range) of the data shown in the growth curve figures.

Response: Thanks for reviewer’s comments. The results of three experiments are presented as individual curves and shown in the revised Figure 2g, 3f, and 6h. There are no additional technical replicates performed in each experiment.

For clarity, the following sentences, “The experiments in (f) and (g) were performed for $n =$

3 biological replicates and each of them is shown”, “The experiments in (e) and (f) were performed for $n = 3$ biological replicates and each of them is shown”, and “The experiments in (g) and (h) were performed for $n = 3$ biological replicates and each of them is shown” have been added in the figure legends of Figure 2g, 3f, and 6h, respectively.

• *Line 178: The authors have made and tested a triple alanine mutant of the AnPycC cyclase. Were single point mutants also tested? Can the authors show that the triple mutant still expresses at a similar level to the wild type or that it is still properly folded?*

Response: The triple alanine mutant of *AnPycC* (F60A/K104A/F109A) was made to test its cUMP production, whereas the single point mutants were not made. The *AnPycC*^{F60A/K104A/F109A} triple mutant could be expressed in high yields and purified to homogeneity as shown below. To further investigate if the purified triple alanine mutant is still properly folded, analytical size-exclusion chromatography (SEC) was applied. The following figure showed that the purified *AnPycC*^{F60A/K104A/F109A} mutant protein eluted as a sharp peak corresponding to monomeric form similar to that of wild-type protein.

Figure. Expression and purification of wild type and triple mutant (F60A/K104A/F109A) of *AnPycC* proteins. (A) 12 % SDS-PAGE analysis of purified *AnPycC*^{F60A/K104A/F109A} proteins. The estimated molecular weight is consistent with its theoretical molecular weight of about 32 kDa. (B) Characterization of *AnPycC*^{WT} and *AnPycC*^{F60A/K104A/F109A} proteins by analytical SEC. Both of them eluted as a sharp peak, corresponding to their monomeric form.

• *Lines 184-186: The strength of this sentence is severely limited because the authors have not shown that UTP is the preferred substrate for AnPycC nor that cUMP is actively produced during the T2 phage infections.*

Response: Thanks for reviewer’s comments. We have further shown that purified *AnPycC* proteins catalyze the conversion of UTP substrates into cUMP products, which have

subsequently been validated by LC-MS/MS experiments. The results have been newly added in Supplementary Fig. 3 and shown as below.

Supplementary Figure 3. Extraction ion chromatography (m/z 305.0 \rightarrow m/z 110.8) and MS/MS spectrum of the LC-MS/MS analysis of (a) the chemical standard of cUMP and (b) cUMP produced in the overnight reaction containing 1 mg/ml *AnPycC* and 1 mM UTP at pH 9.0.

• Lines 270-271: What does excluding a surface area of approximately 1,100 Å² mean in this sentence? It is unclear which interfaces are being referenced and Figure 6a doesn't indicate these buried surfaces.

Response: Thanks for reviewer's comments. To clarify this issue, the clause "excluding a surface area of approximately 1,100 Å²" has been removed and the following sentence "*PsPycTIR* dimer-to-dimer interface comprises interactions between the A-to-A', A-to-B', and B-to-B' protomers (Figure 6b)" has been modified to "*PsPycTIR* dimer-to-dimer interface (~1,100 Å²) comprises interactions between the A-to-A', A-to-B', and B-to-B' protomers (Fig. 7b)".

• Lines 298-299: There may be a typo here, but it is unclear what data is being referred to by this sentence. Is there a terminal residue that was mutated and tested?

Response: Thanks for reviewer's comments. To correct the error, the following modifications have been made:

1. The sentence "Single-point mutation of the N-terminal residue of *PsPycTIR* abolished its NAD⁺ cleavage activity, thus reducing the anti-phage function of *PsPycsar* system" has been removed.
2. The original concluding sentence "In summary, our results support the importance of the N-terminal extension of *PycTIR* proteins for oligomerization and TIR activation" of this paragraph has been replaced by the sentence "In summary, these data demonstrated that

disrupting the key ionic interaction (R12–E114) between N-terminal CNBD domains of *PsPycTIR* dimers severely interfered with the NADase activation of their C-terminal TIR domains and rendered the Pycsar system ineffective for phage defense, supporting the critical role of the N-terminal extension of PycTIR proteins for oligomerization and TIR activation” in lines 320–324.

• *Lines 303-305: The importance of these residues is not validated as no direct binding was measured or observed.*

Response: Thanks for reviewer’s comments. To access the direct involvement of the identified active-site residues of *AnPycC* and *EaPycC* for cUMP and cCMP production, respectively, we have made the active-site alanine mutants of *AnPycC* (F60A/K104A/F109A) and *EaPycC* (F100A/R142A/Q144A) and further performed *in vitro* enzymatic reactions by incubating purified triple alanine mutants *AnPycC*^{F60A/K104A/F109A} and *EaPycC*^{F100A/R142A/Q144A} with UTP and CTP, respectively. The reaction products were then subject to LC-MS/MS analysis. As shown in the following figures, no products were detected in the reaction containing *AnPycC*^{F60A/K104A/F109A} proteins incubated with UTP and *EaPycC*^{F100A/R142A/Q144A} proteins incubated with CTP, supporting the important roles of these residues for cUMP/cCMP production.

To clarify this issue, the sentence “We validated the importance of the residues F60, K104, and F109 on *AnPycC*, and F100, R142, and Q144 from *EaPycC* that directly bind to UTP and CTP” was modified to “... and experimentally verified the important residues F60, K104, and F109 of *AnPycC*, and F100, R142, and Q144 of *EaPycC* for cUMP and cCMP production” in lines 328–330 of the revised manuscript.

Figure. LC-MS/MS analysis of the cyclic pyrimidine products synthesized by PycCs.

(a) Extraction ion chromatography of cCMP signals (m/z 304.0→m/z 109.9) in the LC-MS/MS result of the overnight reaction containing 1 mg/ml *EaPycC*^{F100A/R142A/Q144A} and 1 mM CTP at pH 9.0. The cCMP signal is not observed in the sample. (b) Extraction ion chromatography of cUMP signals (m/z 305.0→m/z 110.8) in the LC-MS/MS analysis of the overnight reaction containing 1 mg/ml *AnPycC*^{F60A/K104A/F109A} and 1 mM UTP at pH 9.0. The cUMP signals is not observed in the sample.

• *Lines 306-308: Cyclic UMP is not a confirmed product of AnPycC and therefore this sentence and the mechanisms of conformational change proposed in this manuscript are not supported by data. The*

experimentally determined structure represents a possible “closed” state of the *AnPycC* cyclase but the “open” state is just a prediction.

Response: Thanks for reviewer’s comments. To validate the production of cUMP by *AnPycC*, we have further performed the enzymatic reaction and demonstrated that purified *AnPycC* proteins are active *in vitro* and catalyze the conversion of UTP into cUMP products. The amounts of cUMP products generated by *AnPycC* increases with increasing pH value of reaction mixture (up to pH 9.0). On the other hand, the inactive conformation of *AnPycC* protein was crystallized in weakly acidic condition (pH 6.5) in this study. Based on these data, it is proposed that *AnPycC* proteins exist in two conformational states: one in inactive conformation and the other in active conformation. The elevated pH value of buffer conditions probably promotes the formation or breakage of some ionic or hydrogen-bonding interactions within PycC cyclases, thereby triggering the conformational transition toward the formation of active conformation of *AnPycC*, which is capable of producing cUMP.

• *Line 321: Is the ratio expected to be 2:2 rather than 1:1 stoichiometry for cUMP if the CNBD resembles that of CRP and is likely a dimer once bound to cyclic nucleotide?*

Response: Thanks for reviewer’s comments. The ratio of cUMP to PycTIR protein is expected to be 2:2, resembling that of CRP dimers.

• *Figure 1: the color scheme is not practical for the superpositions (too many similar overlapping colors)*

Response: Thanks for reviewer’s comments. The color scheme of Figure 1 has been redrawn for easier comparison and shown as below.

Figure 1. Structural characterization of a cCMP-synthesizing PycC cyclase, *EaPycC*.

(c) The front (left) and back (right) view of the superimposed *EaPycC* and *BcPycC* promoters. (d, e) Structural alignment of *EaPycC* dimer with (d) *BcPycC* dimer, and (e) *PaPycC* dimer. *EaPycC*, *BcPycC*, and *PaPycC* proteins are colored in salmon, cyan, and pale-green, with their unique N- and C-terminal structural elements colored in red, blue, and forest-green respectively.

• *Figure 6: Panels C/D use salmon color for one protein and then in panel E of the same figure salmon is used for a different protein (SpHCN). This could be modified for clarity.*

Response: Thanks for reviewer’s comments. We have recolored *SpHCN* protein in original Figure 6e with orange/yellow for clarity. Additionally, protein cartoon representation in original Figure 6e and 6f were changed from “Fancy Helices” to “Cylindrical Helices” for clear comparison. The revised figures were newly added in the revised Fig. 7e and 7f and shown as below.

Figure 7. Molecular mechanism underlying PycTIR oligomerization.

(e) Superimposition of *PsPycTIR*_{CNBD} (cyan) with *SpHCN* (yellow, PDB: 2PTM) with the N-terminal extensions colored in blue and orange, respectively. (f) Superimposition of *PsPycTIR*_{CNBD} (blue-cyan) with *EcCRP* (magenta, PDB: 2CGP).

• *The figure legends (and also the methods sections) describing the NADase assays do not indicate the amount of enzyme used. Additionally, the time point that is being analyzed is not indicated.*

Response: The amount of enzyme used and the time point that is being analyzed in the NADase assays have been added in the methods sections, line 461 and 463 of the revised manuscript, and shown as below:

“Purified full-length *PsPycTIR* proteins (wild-type, R138A, N143A, or N143S mutant) at a final concentration of 1 μ M was then added to start the reaction” and “The changes of relative fluorescence units (RFUs) at the first 15 mins of the reaction were used for analysis”.

For simplicity, this detailed information has been omitted in the figure legends describing the NADase assays.

• *Why was cyclic GMP not tested for NADase stimulation (Figure 5e)?*

Response: Thanks for reviewer's kind reminder. The data for NAD⁺ cleavage activity of *PsPycTIR* in the presence of cyclic GMP at a concentration of 10 μ M has been newly added in the revised Fig. 6e and shown as below.

Figure 6e. NAD⁺ cleavage activity of *PsPycTIR* in the absence or presence of different cyclic mononucleotides at a concentration of 10 μ M.

• *It would be important to show some evidence that the CNBD mutants were still able to be expressed and purified to similar levels as wild type or else data regarding phage defense phenotypes could be inaccurate and actually reflect inactivity of a misfolded protein. Additionally, in support of the oligomerization mechanism proposed for PycTIRs, it would be important to show gels or size exclusion chromatography traces that *NpPycTIR* is a monomer and that *PcPycTIR* is a dimer or that there is some dynamic equilibrium that is influenced by cyclic UMP binding.*

Response: Thanks for reviewer's comments. The single mutant of *PsPycTIR* proteins (N143A and E114R) could be expressed in high yields and purified to homogeneity as shown below. In addition, size exclusion chromatography analysis also demonstrated that both wild-type and single mutant of *PsPycTIR* proteins (N143A and E114R) form folded monomers in solution in the absence of cUMP. Adding cUMP to wild-type *PsPycTIR* protein solutions shifted the equilibrium toward the formation of dimers, tetramers, and large oligomers.

Figure. 12 % SDS-PAGE analysis of purified single mutant (a) *PsPycTIR*^{N143A} and (b) *PsPycTIR*^{E114R} proteins. The estimated molecular weights are slightly larger than their theoretical molecular weight of about 35 kDa.

• Inconsistent use of the abbreviation “LB” – Line 369/370 defined LB as “Luria Bertani broth” and Line 428 defined LB as “lysogeny broth broth”

Response: To be consistently, we keep the definition of LB as “Luria Bertani broth” and changed the word “lysogeny broth broth” to “LB broth” on line 466 of the revised manuscript.

*• Line 400: Need to use specifics when suggesting crystals were cryoprotected. Which crystals were cryoprotected with 15-20% glycerol and which with 10-30% ethylene glycol? The crystallization condition for *EaPycC* already had 25% glycerol and for *PsPycTIR*-CNBD-cUMP already had 6% ethylene glycol and 25% glycerol.*

Response: Thanks for reviewer’s comments. Indeed, the crystals of *EaPycC* and *PsPycTIR*_{CNBD}-cUMP didn’t require additional cryoprotection. The crystals of *AnPycC* and *NpPycTIR*_{CNBD} were cryoprotected with additional 15–20 % glycerol and 10–30 % ethylene glycol, respectively. Therefore, based on the reviewers’ suggestions, we have made the following corrections:

- The sentence “Protein crystals were cryoprotected with reservoir solution supplemented with 15–20 % glycerol or 10–30 % ethylene glycol” has been removed.**
- “...and were cryoprotected with reservoir solution supplemented with 15–20 % glycerol” has been added in the methods section, lines 435–436.**
- “...and were cryoprotected with reservoir solution supplemented with 10–30 % ethylene glycol” has been added in the methods section, lines 438–439.**

Reviewer #2 (Remarks to the Author):

Pycsar (pyrimidine cyclase system for anti-phage resistance) is an antiviral system in bacteria in which a PycC enzyme is activated during phage infection to synthesize a cUMP (3',5'-cyclic uridine monophosphate) or cCMP (3',5'-cyclic cytidine monophosphate) second messenger. A Pycsar effector protein encoded in the same operon recognizes this second messenger through a CNBD (cyclic nucleotide binding domain) and executes abortive infection through either a TIR (Toll/interleukin-1 receptor) NADase domain or membrane-disrupting TM (transmembrane) domains. The mechanism by which PycC enzymes sense phage infection is currently unknown. Furthermore, the molecular details of how Pycsar effectors discriminate between the cNMPs produced by cognate PycC cyclases and “house-keeping” messengers such as cGMP and cAMP remain poorly characterized.

Here, Wang, Yang et al. present crystal structures of the clade E cyclase EaPycC and the clade D cyclase AnPycC. In their structure of AnPycC they identify a zinc-finger motif common to most clade D PycC enzymes and claim that this structural element is required for AnPycC activity during phage infection. Wang, Yang et al. further determine the structure of two CNBDs from PycTIR effector proteins: apo NpPycTIRC�BD and PsPycTIRC�BD in complex with cUMP. Using ITC analysis they demonstrate that several PycTIR proteins recognize cUMP with high affinity. Through their structure of cUMP-bound PsPycTIRC�BD and structure-guided sequence alignment the authors identify the structural determinants of cUMP specificity in CNBDs of PycTIR effectors. Wang, Yang et al. further use their crystal structure of cUMP-bound PsPycTIRC�BD to infer the mechanism of PsPycTIR filament formation upon cUMP recognition.

The structures and analysis presented here do not provide many new insights to the field of Pycsar immunity or bacterial anti-phage defense. Crystal structures of Clade A PaPycC (Linder et al. J. Struc. Biol 2020) and Clade B BcPycC (Tal, Morehouse et al. Cell 2021) have previously been reported. Structural, biochemical, and bioinformatic analysis by Tal, Morehouse et al. previously characterized the conserved active site residues of PycC enzymes and the biochemical requirements for cNMP production. One novel element of this manuscript is the structure and analysis of PsPycTIR bound to cUMP, which provides key insights into specific second messenger recognition by this important family of bacterial immune effectors. Overall, this manuscript requires revisions, including the addition of key biochemical experiments, to be suitable for publication in Nature Communications.

Major Points of Review

- 1. The authors identify key features in the EaPycC and AnPycC crystal structures, including residues presumably involved in UTP and CTP recognition (previously reported) and unique structural features including an N-terminal α 1 helix (previously reported), a “specific Y-shaped structure comprising two anti-parallel β -sheets (β 10– β 13)” in EaPyc and a zinc-ribbon motif in AnPycC.*
- Crystal structures of PaPycC (Linder et al. J. Struc. Biol 2020) and BcPycC (Tal, Morehouse et al.*

Cell 2021) were previously determined. It would significantly improve the clarity of the manuscript and better contextualize the manuscripts findings for the authors to compare (in the “Results” text and the figures) their *EaPycC* and *AnPycC* structures to *PaPycC* and *BcPycC* rather than more distantly related proteins.

Response: We greatly appreciate reviewer’s comments and suggestions. To improve the clarity of the manuscript, structural comparison of *EaPycC* with *PaPycC* (PDB: 6YII) and *BcPycC* (PDB: 7R65) have been newly added in the results sections, lines 150–164 of the revised manuscript and revised Figure 1c–e (see below). Structural comparison of *AnPycC* with *BcPycC*, *PaPycC* and *EaPycC* in original Supplementary Figures 3, 4, and 5 have been combined and moved to the main Figure 4 (see below).

Figure 1. Structural characterization of a cCMP-synthesizing PycC cyclase, *EaPycC*.

(c) The front (left) and back (right) view of the superimposed *EaPycC* and *BcPycC* promoters. The missing residues (a.a. 22–66) of *EaPycC* protomer are indicated by black dashed lines. The inlet shows the enlarged view of the superimposed Y-shaped structure of *EaPycC* with β -hairpin of *BcPycC* at the equivalent position. (d, e) Structural alignment of *EaPycC* dimer with (d) *BcPycC* dimer, and (e) *PaPycC* dimer. *EaPycC*, *BcPycC*, and *PaPycC* proteins are colored in salmon, cyan, and pale-green, with their unique N- and C-terminal structural elements colored in red, blue, and forest-green respectively.

Figure 4. Structural comparison of *AnPycC* with other *PycC* pyrimidine cyclases.

(a) Structural comparison of *AnPycC* protomer with *BcPycC* protomer. (b) Structural comparison of *AnPycC* dimer with *BcPycC* dimer. (c) Left, superimposed *AnPycC* dimer with *PaPycC*. Right, the enlarged view of the superimposed C-terminal structures of *AnPycC* and *PaPycC*. (d) Left, superimposed *AnPycC* dimer with *EaPycC* dimer. Right, the enlarged view of the superimposed C-terminal structures of *AnPycC* and *EaPycC*. *AnPycC*, *BcPycC*, *PaPycC*, and *EaPycC* proteins are colored in pale-green, sky-blue, salmon, and yellow with their unique N- and C-terminal structural elements colored in forest-green, blue, red, and orange respectively. The zinc-coordinating residues of *AnPycC* located at $\alpha 9$ helix and $\beta 10$ - $\alpha 7$ loop are shown in sticks and the bound zinc ions are shown in grey spheres.

- The authors must perform biochemical experiments to analyze the production of cCMP and cUMP by *EaPycC* and *AnPycC* respectively. Previous work in the field (Tal, Morehouse et al.) demonstrates that most *PycC* enzymes are auto-active *in vitro*.

Response: Thanks for reviewer's comments. To experimentally characterize the production of cCMP and cUMP by *EaPycC* and *AnPycC*, respectively, we performed *in vitro* enzymatic reactions by incubating purified *AnPycC* or *EaPycC* proteins with their nucleotide substrates following the procedure previously reported by Tal et. al. (*Cell*. 2021; 184:5728-5739.e16, PMID: 34644530). The reaction mixtures were then subject to LC-MS/MS analysis. The following figures demonstrated that both *EaPycC* and *AnPycC* are auto-active *in vitro* and

synthesize cCMP and cUMP products, respectively. As the reviewer's suggestion, these figures are newly added in the revised manuscript as Supplementary Fig. 1 and 3.

Supplementary Figure 1. Extraction ion chromatography (m/z 304.0 \rightarrow m/z 109.9) and MS/MS spectrum of the LC-MS/MS analysis of (a) the chemical standard of cCMP and (b) cCMP produced in the overnight reaction containing 1 mg/ml *EaPycC* and 1 mM CTP at pH 9.0.

Supplementary Figure 3. Extraction ion chromatography (m/z 305.0 \rightarrow m/z 110.8) and MS/MS spectrum of the LC-MS/MS analysis of (a) the chemical standard of cUMP and (b) cUMP produced in the overnight reaction containing 1 mg/ml *AnPycC* and 1 mM UTP at pH 9.0.

o For all key structures features the authors identify (residues in the catalytic site/ nucleotide binding pocket, the “Y-shaped structure” in *EaPycC*, the zinc-ribbon in *AnPycC*) mutations must be made and

tested for their effect on cNMP synthesis.

o For the authors to claim a role for the zinc-ribbon in AnPycC activity during phage infection, they must investigate its role in cUMP synthesis. If this cannot be done through *in vitro* experiments the authors can use mass-spectrometry to detect cUMP in the lysates of *E. coli* expressing WT or mutated AnPycC during phage infection.

Response: We greatly thanks for reviewer's comments and suggestions.

For key residues in the nucleotide binding/catalytic sites of *EaPycC* and *AnPycC*, we have made triple alanine mutants of *EaPycC* (F100A/R142A/Q144A) and *AnPycC* (F60A/K104A/F109A) and tested their nucleotide cyclase activities through *in vitro* biochemical experiments. The following figures demonstrated that both triple mutants abolished their nucleotide cyclase activities, yielding no cUMP and cCMP products.

Figure. LC-MS/MS analysis of the cyclic pyrimidine products synthesized by PycCs.

(a) Extraction ion chromatography of cCMP signals (m/z 304.0 \rightarrow m/z 109.9) in the LC-MS/MS result of the overnight reaction containing 1 mg/ml *EaPycC*^{F100A/R142A/Q144A} and 1 mM CTP at pH 9.0. The cCMP signal is not observed in the sample. (b) Extraction ion chromatography of cUMP signals (m/z 305.0 \rightarrow m/z 110.8) in the LC-MS/MS analysis of the overnight reaction containing 1 mg/ml *AnPycC*^{F60A/K104A/F109A} and 1 mM UTP at pH 9.0. The cUMP signals is not observed in the sample.

For the zinc-ribbon identified in *AnPycC*, we have made a triple alanine mutant of *AnPycC* (C221A/C223A/C260A). *In vitro* characterization of *AnPycC*^{C221A/C223A/C260A} showed that no cUMP products were detected in the overnight reaction containing *AnPycC*^{C221A/C223A/C260A} mutant proteins and UTP substrates, suggesting the importance of zinc-ribbon for maintaining its uridylyl cyclase activity (see Figure below).

Figure. Extraction ion chromatography of cUMP signals (m/z 305.0 \rightarrow m/z 110.8) in the LC-MS/MS analysis of the overnight reaction containing 1 mg/ml *AnPycC*^{C221A/C223A/C260A} and 1 mM UTP at pH 9.0.

The cUMP signals are not observed in the samples.

The “Y-shaped structure” in *EaPycC* is a unique structure feature among all the solved crystal structures of adenylyl and guanylyl cyclases and PycCs. However, sequence analysis of clade E PycC proteins showed no sequence homology within the region forming Y-shaped structure of *EaPycC*. Therefore, it is considered that the Y-shaped structure of *EaPycC* is a strain-specific adaption of Pycsar defense system in *Elizabethkingia anopheles*. The physiological relevance and exact molecular mechanism regarding this unique structural feature will need further investigation in the future.

- *Crystal structures of EaPycC or AnPycC in complex with CTP/ UTP substrates would add novelty to the manuscript.*

Response: Thanks for reviewer’s suggestions. To obtain complex crystals of *EaPycC*-CTP and *AnPycC*-UTP, we have tried co-crystallization of *EaPycC* and *AnPycC* with different non-hydrolysable CTP and UTP analogues against thousands of crystallization conditions. However, we still cannot get protein crystals of *EaPycC* or *AnPycC* in complex with CTP/UTP after exhausting crystallization condition screenings.

2. *The authors measure the growth of E. coli cells expressing an empty vector or Pycsar systems (WT and mutated) during T2 phage infection (Figs 2g, 3f, 5h).*

- *These experiments are missing critical controls.*

o *The authors must show the growth curves of empty vector cells infected with T2 phage (Figs 2g,3f), as in Fig. 5h.*

o *The authors should also show the growth curves of Pycsar-expressing E. coli in the absence of phage infection (Figs 2g,3f,5h)*

Response: Thanks for reviewer’s comments. The critical control for the growth curves of empty vector cells infected with T2 phage and the growth curves of Pycsar-expressing *E. coli* in the absence of phage infection have been newly added in the results shown in revised Figs. 2g, 3f, and Figs. 2g, 3f, 6h, respectively.

- *The authors should perform an outgrowth of their overnight cultures prior to growth curve measurements in order to ensure each culture is in early log phase at the beginning of the experiment.*

- *The authors must normalize the density of all cultures at the beginning of growth curve measurements.*

In both Fig. 2g and Fig. 3f the cultures start at different OD600 measurements at timepoint 0 and thus it is not possible to compare the growth of different cultures over time.

Response: Thanks for reviewer’s comments. To compare the growth of different cultures over time, we have performed an outgrowth of each overnight culture prior to growth curve

measurements to ensure each culture starts at the same initial OD₆₀₀ in early log phase at the beginning of the experiment. The new results have been updated in revised Fig. 2g, 3f, and 6h and shown as below.

Figure 2g. The growth curves of *E. coli* cells overexpressing wild-type *AnPycC* and *AnPycTM* (orange), *E. coli* cells overexpressing *AnPycC* triple mutant (C221A/C223A/C260A) and *AnPycTM* (purple), and *E. coli* cells harboring empty vector (black) with T2 phage infection at an MOI of 0.1. The *E. coli* cells harboring *AnPycsar* system or empty vector without T2 phage infection and protein induction serve as negative controls.

Figure 3f. The growth curves of *E. coli* cells overexpressing wild-type *AnPycC* and *AnPycTM* (orange), *E. coli* cells overexpressing *AnPycC* triple mutant (F60A/K104A/F109A) and *AnPycTM* (green), and control cells (black) with T2 phage infection at an MOI of 0.1. The *E. coli* cells harboring *AnPycsar* system or empty vector without T2 phage infection and protein induction serve as negative controls.

Figure 6h. The growth curves of *E. coli* cells overexpressing wild-type *PsPysar* system containing *PsPycC* and *PsPycTIR* (red), *E. coli* cells overexpressing *PsPysar* system containing *PsPycC* and *PsPycTIR*^{N143A} (green), *E. coli* cells overexpressing *PsPysar* system containing *PsPycC* and *PsPycTIR*^{E114R} (cyan), and *E. coli* cells carrying empty vector (black) with T2 phage infection at an MOI of 0.01. The *E. coli* cells harboring *PsPysar* system or empty vector without T2 phage infection serve as negative controls.

3. The authors reveal the *PsPycTIRC*NBD dimerization interface in their crystal *PsPycTIRC*NBD – cUMP crystal structure. Based on the crystal packing of *PsPycTIRC*NBD, they further propose that contacts between A' and A, A and B', and B' and B protomers in adjacent dimers mediate *PsPycTIR* linear filament formation.

- It was previously shown by electron microscopy that *PycTIR* proteins form filaments specifically in the presence of their cognate cNMP ligand (Tal, Morehouse, et al., Cell 2021).

- The crystal structure contains only the CNBD and not the TIR domain of *PsPycTIR*, and thus it is not clear that this structure provides relevant insight into filament formation of the full-length protein. Previous structures of nucleotide-bound *SAVED-TIR* and *STING-TIR* proteins reveal important TIR-TIR contacts involved in filament formation (Hogrel et al., Nature 2022; Morehouse et al., Nature 2022).

- To demonstrate the relevance of the proposed A' and A, A and B', and B' and B interfaces for filament formation, the authors should analyze WT vs. interface mutant (e.g. E114R) *PsPycTIR* oligomerization using an *in vitro* assay such as SEC-MALS.

Response: Thanks for reviewer's comments. There is no such high value instrument in my school. In my country, SEC-MALS instruments are owned by Academia Sinica, which are not available for loan. Instead, we utilized analytical gel-filtration to demonstrate the relevance of the proposed oligomerization interfaces for *PsPycTIR* filament formation. As shown in the following figure, binding of cUMP induces the formation of large oligomers of *PsPycTIR*^{WT} that probably represent long filaments. In contrast, the oligomerization mutant *PsPycTIR*^{E114R} forms only dimers and tetramers even in the presence of excessive cUMP, validating the importance of the observed ionic interaction (E114-R12) on the A' and A and B' and B interfaces for *PsPycTIR*

filament formation.

Figure. SEC profiles of full-length *PsPycTIR*^{WT} (blue line) and oligomerization interface mutant *PsPycTIR*^{E114R} (red line) in the presence of cUMP at 2-fold molar excess.

Minor Points of Review

1. The authors speculate that “the ligand-free *AnPycC* presented here is in its inactive state with closed catalytic centers and may be activated by the binding of phage proteins” (lines 171-173). However the authors’ data also demonstrates that when heterologously expressed in *E. coli*, *AnPycC* WT and its effector *AnPycTM* are extremely toxic (Fig. 2f, 3e), suggesting auto-activity by the WT *AnPyc* system in the absence of phage proteins.

- To resolve whether *AnPycC* is active in the absence of phage proteins, the authors should test the activity of the purified *AnPycC* protein *in vitro* (see major point 1).

Response: Thanks for reviewer’s comments. As responded in major point 1, the cUMP has been detected as the product of the *in vitro* experiment of *AnPycC* and UTP in the absence of phage proteins. We further performed the activity assay of the purified *AnPycC* protein in the presence of phage proteins *in vitro*; however, cUMP was not detected (as shown in the following figure). The possible reason may be that in the presence of the phage proteins in the *in vitro* system, abundant phage protein mixture may hinder the contact between *AnPycC* and UTP and results in no production of cUMP.

Figure. Extraction ion chromatography (m/z 305.1→m/z 111.0) and MS/MS spectrum of the LC-MS/MS

analysis of (a) the chemical standard of cUMP and (b) the product solution of the reaction of 1 mg/ml *AnPycC*, phage proteins and 1 mM UTP at pH 9. The cUMP signal is not observed in the product solution.

- It is intriguing that the AlphaFold model of *AnPycC* is in an “open”, auto-active form in contrast to the “closed” crystal structure of *AnPycC*. Is it possible that one of the co-purifying molecules in the *AnPycC* crystal is inducing the “closed” conformation, potentially by mimicking an inhibitory ligand?

Response: Thanks for reviewer’s comment. In the *AnPycC* crystal, it is found that two molecules of glycerol occupied the two symmetric active sites of a dimeric *AnPycC* protein. Thus, it is possible that the co-purifying glycerol molecules in the *AnPycC* crystal induced the formation of the inactive conformation, which exclude the binding of UTP substrates. However, the detailed molecular mechanism underlying how glycerol or glycerol-like molecules affect the structure of *AnPycC* will need further experimental characterization in the future.

2. It might be helpful if the authors made a graph to summarize the K_D of different *PycTIR* proteins for Fig. 4a, as measured by ITC (Supp Fig. 6), in place of the ITC analysis for only *PsPycTIR* that is currently shown in Fig. 4a.

Response: Thanks for reviewer’s comments. We have summarized the K_D values of different *PycTIR* proteins in a table in place of the ITC analysis for only *PsPycTIR*. The new Fig. 5a has been updated and shown as below.

	K_D (M)
PsPycTIR _{CNBD}	1.0×10^{-9}
NpPycTIR _{CNBD}	1.0×10^{-9}
AhPycTIR _{CNBD}	8.2×10^{-7}
SmPycTIR _{CNBD}	2.7×10^{-7}

Figure 5a. A summary of dissociation constants of different *PycTIR* proteins for cUMP as determined by ITC.

- It would be interesting, but not essential, for the authors to test the affinity of *PycTIR* proteins for other cyclic mononucleotides in comparison to cUMP to demonstrate specificity of these effectors for their cognate second messenger.

Response: Thanks for reviewer’s comments. Cyclic mononucleotides, including cAMP, cGMP, and cCMP, have been tested for their binding to *PsPycTIR* proteins. The titration results showed that the binding of cAMP, cGMP, or cCMP to *PsPycTIR*, was too weak to be detected using ITC instruments.

3. The most novel element of this manuscript is the identification of residues which control specific recognition of cNMP messengers by PycTIR effectors. The authors could provide further experiments related to this finding to enhance their manuscript.

- The authors could use ITC to analyze the affinity of PsPycTIR cUMP-binding mutants for cUMP and other cNMPs. The authors could make analogous mutations in the other PycTIR proteins tested in this manuscript to demonstrate the conserved importance of these residues for cNMP recognition.

Response: Thanks for the reviewer's comment. To validate the cUMP-recognizing residues of PsPycTIR observed in crystal structure reported in this manuscript, we have performed NADase activity assays, protein toxicity analysis in *E. coli*, and phage infection assays to test PsPycTIR cUMP-binding mutants. In vitro NADase activity assays demonstrated that mutating either base-recognition residue N143 or base-stacking residue R138 of PsPycTIR to alanine or serine totally abolished their responsiveness to cUMP (Fig. 6f). In vivo protein toxicity analysis in *E. coli*, and phage infection assays further validated the importance of cUMP-binding residues N143 and R138 of PsPycTIR for activation of effector function to combat against phage infections (Fig. 6g, h).

4. T2 phage infection of cells expressing a PsPycTIR cUMP-recognition mutant and putative oligomerization mutant, which should be unable to be activated for NADase activity, do not appear significantly more susceptible to phage infection (Fig. 5h). The authors should address this.

Response: Thanks for the reviewer's comment. Due to the critical issues raised by reviewer in major point 2, we have modified the protocol for phage infection experiments according to the reviewer's suggestions and redone the experiments. The new results have been updated in revised Fig. 6h and shown as below. The resulting curves demonstrated that *E. coli* cells harboring PsPycsar system with either cUMP-recognition mutant PsPycTIR^{N143A} or oligomerization mutant PsPycTIR^{E114R} are significantly more susceptible to phage infection than *E. coli* cells expressing wild-type system.

Figure 6h. The growth curves of *E. coli* cells overexpressing wild-type PsPycsar system containing PsPycC

and PsPycTIR (red), *E. coli* cells overexpressing PsPysar system containing PsPycC and PsPycTIR^{N143A} (green), *E. coli* cells overexpressing PsPysar system containing PsPycC and PsPycTIR^{E114R} (cyan), and *E. coli* cells carrying empty vector (black) with T2 phage infection at an MOI of 0.01.

5. Text comments

- The authors should include the residue numbers for each protein in the sequence alignment in Fig. 5c.

Response: Thanks for the reviewer's comment. The residue numbers for each protein in the sequence alignment has been newly added. The revised figure has been updated in new Fig. 6c and shown as below.

Figure 6c. Sequence alignment of CRP proteins with PycTIR proteins. The conserved residues interacting with cyclic mononucleotides are indicated by triangles (▼). The specificity-determining residue of PycTIR proteins and CRP proteins is indicated by an asterisk (*).

- Line 223: “invisible” should be written as “unresolved in the NpPycTIRCNBD crystal structure”.
- Line 669: “regulate” should be corrected to “regulates”.
- Line 798: “yelloworange” should be corrected to “yellow-orange”.

Response: Thanks for reviewer's comments. These errors have been corrected accordingly in the revised manuscript.

- The authors should reference the following paper for use of AlphaFold structural predictions: Jumper et al. Nature 2021

Response: Thanks for reviewer's comments. The reference has been added in line 201 (ref. no. 23) of the revised manuscript.

REVIEWER COMMENTS

Reviewer #1 (Remarks to the Author):

Yang et al. made several important modifications to the manuscript in response to the reviewers. Their introduction now emphasizes the importance of pyrimidine nucleotides in bacterial defense systems against phage infections. They also included descriptions of the different clades of PycC cyclases and provided structural comparisons between various Pycsar cyclases with EaPycC. LC-MS/MS analysis was conducted to confirm the production of cyclic CMP and cyclic UMP by EaPycC and AnPycC, respectively. Site-directed mutagenesis was added to validate the importance of specific residues for substrate recognition and catalysis. The authors now included proper controls, including empty vector controls, to provide a baseline for phage infection without a Pycsar defense system. Growth curves were adjusted to ensure consistent starting OD600 values for all conditions. The authors provided additional methodological details, such as the amount of enzyme used in assays, and the time points analyzed. Color schemes were adjusted for clarity in figures, and legends were updated to provide clearer explanations of the data presented. Overall, Yang et al. addressed many of the concerns raised by both reviewers by providing additional experimental data, clarifications, and improvements to the manuscript's presentation.

In spite of the new results (which do strengthen the manuscript), several points of concern still remain- in particular regarding possible over-interpretation of the results. Oddly, some of the data which would further strengthen the story appear absent from the new supplementary data figures despite being presented in the response to reviewer comments. This should be a relatively simple fix. However, data such as chromatography and gel analysis which should have been presented, are still missing. More descriptive suggestions are provided below.

- Line 149 missing words “structures of” following “determined”
- Line 165-166: The results reported do not provide a structural basis for differentiation between cCMP- and cUMP-synthesizing PycCs. No evidence was provided to show that EaPycC doesn't use UTP as a substrate, nor were the structural differences between EaPycC and other reported PycC homologs shown to be essential for discrimination between the two pyrimidine substrates. Suggestion is to remove this line of text (as it is unsupported by data), to otherwise soften the language, or to test alternative substrate preference for EaPycC and other PycC homologs tested in this manuscript.
- Gels (SDS-PAGE) and/or size exclusion profiles for the Zn²⁺ binding triple mutant need to be provided to show that protein stability/dimerization has not been impacted by this drastic mutation which the authors interpret as being essential for phage defense (rather than folding). The size exclusion chromatography traces and SDS-PAGE analysis for all mutants should be provided in the supplementary information. Some of these data were presented in the rebuttal however critical information regarding the proper folding and oligomerization of the zinc binding ribbon mutant (in particular, but also the E114R mutant without cUMP addition) is still missing. In addition, the analysis regarding the oligomeric state of AnPycC is confusing- All adenylate/guanylate/PycC are

known to operate as dimers and indeed the structural data from this manuscript supports this feature of AnPycC however the authors have indicated that this protein is a monomer in solution based on size exclusion profile.

- Line 294: “Here, we found that PsPycTIR can potentially oligomerize.” Why potentially? Wasn’t this explicitly shown using size exclusion chromatography in the rebuttal? The discussion regarding oligomeric state of PycTIRs needs support from data and text changes to reflect that there is experimental evidence in addition to speculation regarding interfaces observed in the crystal structures.
- Line 336: The in vitro enzyme activity assay only shows that mutation of the zinc-binding ribbon of AcPycC can disrupt cUMP synthesis. It does not, however, prove that the zinc-binding ribbon is required to stabilize the active-state conformation that can bind UTP. I suggest changing the language to reflect the limitations of what the data tell us about mechanism (nothing about the active or inactive conformation can be known) or doing additional experiments to verify that the mutant 1) is still properly folded 2) can or cannot bind UTP.
- ITC data supporting a lack of direct binding to other cyclic nucleotides which were tested (according to the rebuttal) should be provided in the supplementary information as these data are currently not shown.
- Data regarding enzymatic activity of the cyclases at various pH levels should be included if possible (data not shown but referenced in the rebuttal). These data would provide evidence for the mechanistic interpretation of active vs inactive structures potentially being influenced by pH.

Reviewer #2 (Remarks to the Author):

This revised manuscript has sufficiently addressed nearly all previous comments and is significantly improved from the original submission. It is suitable for publication in Nature Communications and helps to advance our molecular understanding of nucleotide second messenger immune signaling in bacteria.

Comparative structural analysis of the new EaPycC and AnPycC structures determined by the authors and the previously determined BcPycC and PaPycC structures greatly clarifies the significance of these new structures. The alignment of EaPycC and BcPycC active site residues (Supp. Fig. 2d) reveals the structural determinants for CTP substrate specificity in Clade E Pycsar cyclases—an important insight for the field.

The LC-MS/MS data demonstrating EaPycC and AnPycC cCMP and cUMP synthesis in vitro is also a nice addition to the manuscript; however, the in vitro assays showing a lack of activity for the EaPycC and AnPycC catalytic mutants and the AnPycC zinc-ribbon mutant might also be added to the supplementary figures.

Additionally, the growth curve assays are greatly improved from the original manuscript and more convincingly support the authors' conclusions. However, in Figure 2f the authors should plot the

growth curve of the uninduced AnPycC C221A/C223A/C260A culture as a control. Additionally, in Figure 2f there only appears to be 2 of 3 biological replicates shown.

In line 295 the authors should clarify that the PsPcyTIR dimers they are discussing are only the CNBD. While the dimer-dimer interface they analyze is most likely biologically important, it is important to note that the TIR domain is absent from the PsPcyTIR crystal and thus the authors cannot analyze TIR-TIR interfaces that might also be important for filament formation.

Finally, while it is helpful to contextualize Pycsar defense systems in terms of nucleotide signal-mediated bacterial immunity, the placement of the sentence in lines 84 – 86 (“Purine nucleotide signals ... several decades”) is confusing. The sentence order implies that cAMP and cGMP are known as bacterial immune signals; however, these molecules have only been characterized as second messengers in non-immune signaling pathways.

Author Rebuttals

We express our sincere gratitude to the reviewers' constructive comments to improve our manuscript. We have carefully addressed all the issues raised by the reviewers on a point-by-point basis below and revised the manuscript according to the reviewers' comments.

Reviewer #1 (Remarks to the Author):

Yang et al. made several important modifications to the manuscript in response to the reviewers. Their introduction now emphasizes the importance of pyrimidine nucleotides in bacterial defense systems against phage infections. They also included descriptions of the different clades of PycC cyclases and provided structural comparisons between various Pycsar cyclases with EaPycC. LC-MS/MS analysis was conducted to confirm the production of cyclic CMP and cyclic UMP by EaPycC and AnPycC, respectively. Site-directed mutagenesis was added to validate the importance of specific residues for substrate recognition and catalysis. The authors now included proper controls, including empty vector controls, to provide a baseline for phage infection without a Pycsar defense system. Growth curves were adjusted to ensure consistent starting OD600 values for all conditions. The authors provided additional methodological details, such as the amount of enzyme used in assays, and the time points analyzed. Color schemes were adjusted for clarity in figures, and legends were updated to provide clearer explanations of the data presented. Overall, Yang et al. addressed many of the concerns raised by both reviewers by providing additional experimental data, clarifications, and improvements to the manuscript's presentation.

In spite of the new results (which do strengthen the manuscript), several points of concern still remain in particular regarding possible over-interpretation of the results. Oddly, some of the data which would further strengthen the story appear absent from the new supplementary data figures despite being presented in the response to reviewer comments. This should be a relatively simple fix. However, data such as chromatography and gel analysis which should have been presented, are still missing. More descriptive suggestions are provided below.

- *Line 149 missing words “structures of” following “determined”*

Response: The original sentence has been modified to “...determined structures of a clade A PycC from *Pseudomonas aeruginosa* (PaPycC, PDB: 6YII) and a clade B PycC...” according to the reviewer's suggestion.

- *Line 165-166: The results reported do not provide a structural basis for differentiation between cCMP- and cUMP-synthesizing PycCs. No evidence was provided to show that EaPycC doesn't use UTP as a substrate, nor were the structural differences between EaPycC and other reported PycC*

homologs shown to be essential for discrimination between the two pyrimidine substrates. Suggestion is to remove this line of text (as it is unsupported by data), to otherwise soften the language, or to test alternative substrate preference for *EaPycC* and other *PycC* homologs tested in this manuscript.

Response: Thanks for reviewer’s comments. We have deleted the sentence “Altogether, these results provide structural basis for differentiation between cCMP- and cUMP-synthesizing *PycCs*” according to the reviewer’s suggestion.

• Gels (SDS-PAGE) and/or size exclusion profiles for the Zn^{2+} binding triple mutant need to be provided to show that protein stability/dimerization has not been impacted by this drastic mutation which the authors interpret as being essential for phage defense (rather than folding). The size exclusion chromatography traces and SDS-PAGE analysis for all mutants should be provided in the supplementary information. Some of these data were presented in the rebuttal however critical information regarding the proper folding and oligomerization of the zinc binding ribbon mutant (in particular, but also the E114R mutant without cUMP addition) is still missing.

Response: We greatly thanks for reviewer’s suggestions. SDS-PAGE and size exclusion chromatography profile for *AnPycC* Zn^{2+} binding triple mutant have been newly provided in Supplementary Fig. 10a and 10d, respectively, to show that protein stability and oligomeric state have not been impacted by this drastic mutation. We have now included size exclusion chromatography traces and SDS-PAGE analysis for protein variants in Supplementary Fig. 10. Particularly, SEC analysis of *PsPycTIR* E114R mutant without cUMP addition has been newly added to Supplementary Fig. 10g.

Supplementary Figure 10. SDS-PAGE and size exclusion chromatography profiles for all mutant proteins. (a) 12 % SDS-PAGE analysis of purified *AnPycC*^{WT}, *AnPycC*^{F60A/K104A/F109A}, *AnPycC*^{C221A/C223A/C260A}, *PsPycTIR*^{WT}, *PsPycTIR*^{E114R}, and *PsPycTIR*^{N143A}. Each protein was purified to >95% purity. (b–i) Analytical size exclusion chromatography of (b) *AnPycC*^{WT}, (c) *AnPycC*^{F60A/K104A/F109A}, (d) *AnPycC*^{C221A/C223A/C260A}, (e) *PsPycTIR*^{WT}, (f) *PsPycTIR*^{WT} with cUMP, (g) *PsPycTIR*^{E114R}, (h) *PsPycTIR*^{E114R} with cUMP, and (i) *PsPycTIR*^{N143A}. These SEC profiles indicate the proper folding and oligomerization of the purified proteins.

In addition, the analysis regarding the oligomeric state of AnPycC is confusing- All adenylate/guanylate/PycC are known to operate as dimers and indeed the structural data from this manuscript supports this feature of AnPycC however the authors have indicated that this protein is a monomer in solution based on size exclusion profile.

Response: Thanks for reviewer’s comments. After the literature search, we found that many of the adenylate and guanylate cyclases have been reported to exist in an equilibrium mixture of monomers as major form and dimers as minor form in solution (Barathy, et al., IUCrJ. 2014, PMID: 25295175; Kumar, et al., J Biol Chem. 2017, PMID: 29118188; Vercellino, et al., PNAS. 2017, PMID: 29087332). However, the related analysis of PycC cyclases are currently lacking. In this study, the purified *AnPycC* proteins all eluted as monomers in solution based on analytical size exclusion chromatography. On the other hand, the crystal structure of *AnPycC* do show dimeric architecture. Thus, the exact molecular mechanisms underlying formation of monomer/dimer of *AnPycC* is still ambiguous and will need further studies and investigation in the future.

• *Line 294: “Here, we found that PsPycTIR can potentially oligomerize.” Why potentially? Wasn’t this explicitly shown using size exclusion chromatography in the rebuttal? The discussion regarding oligomeric state of PycTIRs needs support from data and text changes to reflect that there is experimental evidence in addition to speculation regarding interfaces observed in the crystal structures.*

Response: Thanks for reviewer’s comments. The original sentence has been modified to “Here, we found that *PsPycTIR* can oligomerize.” The descriptions regarding oligomeric state of *PsPycTIRs* have all been modified to reflect that there is experimental evidence rather than speculation.

• *Line 336: The in vitro enzyme activity assay only shows that mutation of the zinc-binding ribbon of AcPycC can disrupt cUMP synthesis. It does not, however, prove that the zinc-binding ribbon is required to stabilize the active-state conformation that can bind UTP. I suggest changing the language to reflect the limitations of what the data tell us about mechanism (nothing about the active or inactive conformation can be known) or doing additional experiments to verify that the mutant 1) is still*

properly folded 2) can or cannot bind UTP.

Response: Thanks for reviewer's comments. To reflect the limitations of the current data, we have softened the language by changing the original sentence "...it is suggested that the zinc-binding ribbon of *AnPycC* is required to stabilize the active-state conformation that can bind UTP substrates and catalyze the synthesis of cUMP" to "...it is suggested that the zinc-binding ribbon of *AnPycC* may stabilize certain structural element that is associated with production of cUMP" in line 341 of the revised manuscript.

• *ITC data supporting a lack of direct binding to other cyclic nucleotides which were tested (according to the rebuttal) should be provided in the supplementary information as these data are currently not shown.*

Response: The ITC instrument in our lab is out of action due to the strong earthquakes in Taiwan and is currently under maintenance. Because reviewer's opinions are very important, we switched to currently available alternatives. Instead, we utilize protein thermal shift assay to test the ability of cyclic mononucleotides to stabilize *PsPycTIR* proteins. As shown in the following figure, cUMP caused the most significant shift of 21 °C in comparison to other cyclic mononucleotides. As the reviewer's suggestion, these figures are newly added in the revised manuscript as Supplementary Fig. 6.

Supplementary Figure 6. Thermal shift assay for cyclic nucleotides binding to the CNBD domain of *PsPycTIR* proteins.

Thermal shift analysis of *PsPycTIR*_{CNBD} in the absence (black line) or presence of cAMP (red), cGMP (yellow), cUMP (cyan), and cCMP (green).

• *Data regarding enzymatic activity of the cyclases at various pH levels should be included if possible (data not shown but referenced in the rebuttal). These data would provide evidence for the mechanistic interpretation of active vs inactive structures potentially being influenced by pH.*

Response: Thanks for reviewer's comments. The data regarding *in vitro* enzymatic reactions by incubating purified *AnPycC* proteins with UTP at various pH values, including 6.5, 8.0, and 9.0, have now presented in Supplementary Fig. 3b–3d and shown below.

Supplementary Figure 3. Extraction ion chromatography (m/z 305.0 \rightarrow m/z 110.8) and MS/MS spectrum of the LC-MS/MS analysis of the cUMP products synthesized by *AnPycC*.

(a) LC-MS/MS analysis of the chemical standard of cUMP. (b–d) LC-MS/MS analysis of cUMP produced in the overnight reaction containing 1 mg/ml *AnPycC*^{WT}, 1 mM UTP, 1 mM Mg²⁺, and 1 mM Mn²⁺ at (b) pH 9.0, (c) pH 8.0, and (d) pH 6.5. (e, f) LC-MS/MS analysis of cUMP produced in the overnight reaction containing 1 mg/ml (e) *AnPycC*^{C221A/C223A/C260A} or (f) *AnPycC*^{F60A/K104A/F109A} with 1 mM UTP, 1 mM Mg²⁺, and 1 mM Mn²⁺ at pH 9.0. The cUMP signals are not observed in (c–f).

Reviewer #2 (Remarks to the Author):

This revised manuscript has sufficiently addressed nearly all previous comments and is significantly improved from the original submission. It is suitable for publication in Nature Communications and helps to advance our molecular understanding of nucleotide second messenger immune signaling in bacteria.

Comparative structural analysis of the new EaPycC and AnPycC structures determined by the authors and the previously determined BcPycC and PaPycC structures greatly clarifies the significance of these new structures. The alignment of EaPycC and BcPycC active site residues (Supp. Fig. 2d) reveals the structural determinants for CTP substrate specificity in Clade E PycC cyclases—an important insight for the field.

The LC-MS/MS data demonstrating EaPycC and AnPycC cCMP and cUMP synthesis in vitro is also a nice addition to the manuscript; however, the in vitro assays showing a lack of activity for the EaPycC and AnPycC catalytic mutants and the AnPycC zinc-ribbon mutant might also be added to the supplementary figures.

Response: Thanks for reviewer's comments. The results of the biochemical experiments to analyze the production of cCMP and/or cUMP by *EaPycC* and *AnPycC* catalytic mutants and the *AnPycC* zinc-ribbon mutant have now been included in Supplementary Fig. 1c and 3e–3f.

Supplementary Figure 1. Extraction ion chromatography (m/z 304.0 \rightarrow m/z 109.9) and MS/MS spectrum of LC-MS/MS analysis of the cCMP products synthesized by *EaPycC*.

(c) LC-MS/MS analysis of cCMP produced in the overnight reaction containing 1 mg/ml *EaPycC*^{F100A/R142A/Q144A} with 1 mM CTP, 1 mM Mg²⁺, and 1 mM Mn²⁺ at pH 9.0. The cCMP signal is not observed.

Supplementary Figure 3. Extraction ion chromatography (m/z 305.0 \rightarrow m/z 110.8) and MS/MS spectrum of the LC-MS/MS analysis of the cUMP products synthesized by *AnPycC*.

(e, f) LC-MS/MS analysis of cUMP produced in the overnight reaction containing 1 mg/ml (e) *AnPycC*^{C221A/C223A/C260A} or (f) *AnPycC*^{F60A/K104A/F109A} with 1 mM UTP, 1 mM Mg²⁺, and 1 mM Mn²⁺ at pH 9.0. The cUMP signals are not observed in (e–f).

*Additionally, the growth curve assays are greatly improved from the original manuscript and more convincingly support the authors conclusions. However, in Figure 2f the authors should plot the growth curve of the uninduced *AnPycC* C221A/C223A/C260A culture as a control. Additionally, in Figure 2f there only appears to be 2 of 3 biological replicates shown.*

Response: Thanks for reviewer’s comments. The growth curves of the uninduced *E. coli* culture harboring *AnPycC*^{C221A/C223A/C260A} have been newly added to Figure 2f as a control and shown below. In Figure 2f, each of the three biological replicates have been shown. To make it more clear, the line width of each replicate has been reduced to 0.75 pt.

Figure 2. Clade D PycC contains a conserved zinc-finger, which regulates uridylate cyclase and anti-phage ability

(f) The growth curves of *E. coli* cells overexpressing wild-type *AnPycC* and its effector *AnPycTM* (orange lines) and *E. coli* cells overexpressing *AnPycC* triple mutant (C221A/C223A/C260A) and *AnPycTM* (purple lines) compared with uninduced control cells (dashed lines).

*In line 295 the authors should clarify that the *PsPcyTIR* dimers they are discussing are only the CNBD. While the dimer-dimer interface they analyze is most likely biologically important, it is important to note that the TIR domain is absent from the *PsPcyTIR* crystal and thus the authors cannot analyze TIR-TIR interfaces that might also be important for filament formation.*

Response: Thanks for reviewer's comments. We have replaced *PsPcyTIR* with *PsPcyTIR_{CNBD}* in lines 296, 297, 307, 310, 312, and 315 to clarify this issue. Furthermore, we have added an additional comment "Despite these important findings, it should be noted that the TIR domain is absent from the complex structure of *PsPcyTIR_{CNBD}*-cUMP and thus the TIR-TIR interfaces that might also be important for filament formation cannot be analyzed and will need further investigation in the future" in lines 325–328 according to the reviewer's suggestion.

*Finally, while it is helpful to contextualize *Pycsar* defense systems in terms of nucleotide signal-mediated bacterial immunity, the placement of the sentence in lines 84 – 86 ("Purine nucleotide signals ... several decades") is confusing. The sentence order implies that cAMP and cGMP are known as bacterial immune signals; however, these molecules have only been characterized as second messengers in non-immune signaling pathways.*

Response: Thanks for reviewer's comments. To avoid confusing, the sentence has been changed to "Purine nucleotide signals have been subject to research for decades." in line 84 of the revised manuscript.

REVIEWERS' COMMENTS

Reviewer #1 (Remarks to the Author):

The additional data and text edits are important contributions to the manuscript. In general, I feel that the main claims are well supported now with evidence. I would only recommend that the authors consider providing the following points of analysis in their discussion to round out the study in lieu of collecting more data: Adenylate/guanylate synthases isolated as monomers which undergo oligomerization to dimers as related to the activation mechanism is a concept which could be expanded on in the discussion. The authors have provided the foundations for this argument in their rebuttal response to reviewer comments including citation of several publications in support of their discovery that AnPycC is a monomer in vitro and a dimer in the crystals.

Reviewer #2 (Remarks to the Author):

The authors have sufficiently addressed the major points brought up by both reviewers and the manuscript has further improved from the revised submission. The inclusion of new data in the supplementary figures is an important addition to complete the manuscript. I support the publication of this revised manuscript in Nature Communications.

Author Rebuttals

Reviewer #1 (Remarks to the Author):

The additional data and text edits are important contributions to the manuscript. In general, I feel that the main claims are well supported now with evidence. I would only recommend that the authors consider providing the following points of analysis in their discussion to round out the study in lieu of collecting more data: Adenylate/guanylate synthases isolated as monomers which undergo oligomerization to dimers as related to the activation mechanism is a concept which could be expanded on in the discussion. The authors have provided the foundations for this argument in their rebuttal response to reviewer comments including citation of several publications in support of their discovery that AnPycC is a monomer in vitro and a dimer in the crystals.

Response: Thanks for reviewer's comments and suggestions. We have added the following descriptions to support our discovery in the discussion section, lines 358–363:

“In addition to the pH value and phage components, there may be other factors that can influence the cyclase activity of PycCs. It is reported that adenylate and guanylate cyclases isolated as monomers can undergo oligomerization to dimers to become active³³⁻³⁵. In this study, we found that purified AnPycC proteins eluted as monomers in solution but form dimers in the crystals (Supplementary Fig. 10b). Whether oligomerization of PycC cyclases acts as another activation mechanism will need further studies and investigation in the future.”

References:

- 33 Barathy, D., Mattoo, R., Visweswariah, S. & Suguna, K. New structural forms of a mycobacterial adenylyl cyclase Rv1625c. *IUCrJ* 1, 338-348, (2014).**
- 34 Kumar, R. P. *et al.* Structure and monomer/dimer equilibrium for the guanylyl cyclase domain of the optogenetics protein RhoGC. *J Biol Chem* 292, 21578-21589, (2017).**
- 35 Vercellino, I. *et al.* Role of the nucleotidyl cyclase helical domain in catalytically active dimer formation. *Proc Natl Acad Sci U S A* 114, E9821-e9828, (2017).**

Reviewer #2 (Remarks to the Author):

The authors have sufficiently addressed the major points brought up by both reviewers and the manuscript has further improved from the revised submission. The inclusion of new data in the supplementary figures is an important addition to complete the manuscript. I support the publication of this revised manuscript in Nature Communications.

Response: We express our gratitude to the reviewer's invaluable and insightful comments and suggestions to improve our manuscript.